# A Scaling Law for Syn2real Transfer: How Much Is Your Pre-training Effective?

## Abstract

Synthetic-to-real transfer learning is a framework in which a synthetically generated dataset is used to pre-train a model to improve its performance on real vision tasks. The most significant advantage of using synthetic images is that the ground-truth labels are automatically available, enabling unlimited expansion of the data size without human cost. However, synthetic data may have a huge domain gap, in which case increasing the data size does not improve the performance. How can we know that? In this study, we derive a simple scaling law that predicts the performance from the amount of pre-training data. By estimating the parameters of the law, we can judge whether we should increase the data or change the setting of image synthesis. Further, we analyze the theory of transfer learning by considering learning dynamics and confirm that the derived generalization bound is consistent with our empirical findings. We empirically validated our scaling law on various experimental settings of benchmark tasks, model sizes, and complexities of synthetic images.

## 1 Introduction

The success of deep learning relies on the availability of large data. If the target task provides limited data, the framework of transfer learning is preferably employed. A typical scenario of transfer learning is to pre-train a model for a similar or even different task and fine-tune the model for the target task. However, the limitation of labeled data has been the main bottleneck of supervised pre-training. While there have been significant advances in the representation capability of the models and computational capabilities of the hardware, the size and the diversity of the baseline dataset have not been growing as fast (Sun et al., 2017). This is partially because of the sheer physical difficulty of collecting large datasets from real environments (e.g., the cost of human annotation).

In computer vision, *synthetic-to-real (syn2real) transfer* is a promising strategy that has been attracting attention (Su et al., 2015; Movshovitz-Attias et al., 2016; Georgakis et al., 2017; Tremblay et al., 2018; Hinterstoisser et al., 2019; Borrego et al., 2018; Chen et al., 2021). In syn2real, images used for pre-training are synthesized to improve the performance on real vision tasks. By combining various conditions, such as 3D models, textures, light conditions, and camera poses, we can synthesize an infinite number of images with ground-truth annotations. Syn2real transfer has already been applied in some real-world applications. Teed & Deng (2021) proposed a simultaneous localization and mapping (SLAM) system that was trained only with synthetic data and demonstrated state-of-the-art performance. The object detection networks for autonomous driving developed by Tesla was trained with 370 million images generated by simulation (Karpathy, 2021).

The performance of syn2real transfer depends on the similarity between synthetic and real data. In general, the more similar they are, the stronger the effect of pre-training will be. On the contrary, if there is a significant gap, increasing the number of synthetic data may be completely useless, in which case we waste time and computational resources. A distinctive feature of syn2real is that we can control the process of generating data by ourselves. If a considerable gap exists, we can try to regenerate the data with a different setting. But how do we know that? More specifically, in a standard learning setting without transfer, a "power law"-like relationship called a *scaling law* often holds between data size and generalization errors (Rosenfeld et al., 2019; Kaplan et al., 2020). Is there such a rule for pre-training?

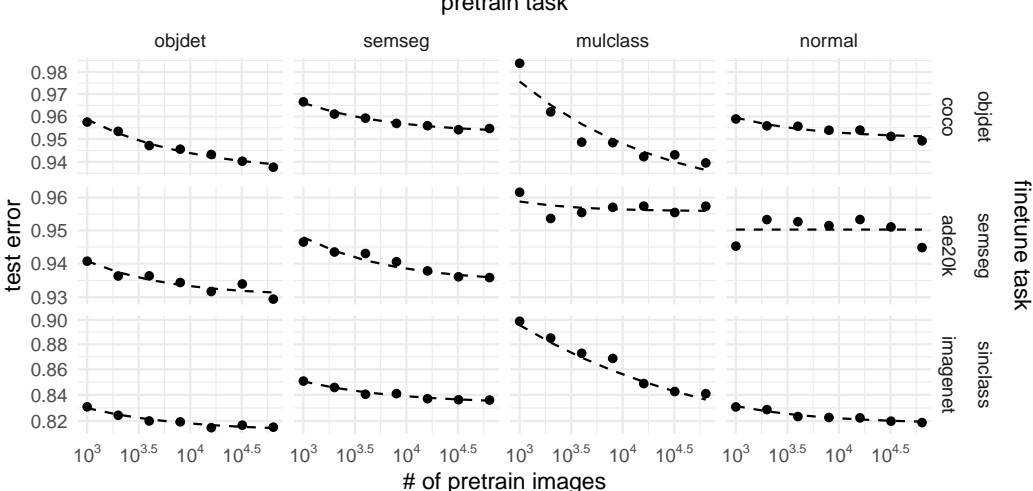

Figure 1: Empirical results of syn2real transfer for different tasks. We conducted four pre-training tasks: object detection (`objdet`), semantic segmentation (`semseg`), multi-label classification (`mulclass`), surface normal estimation (`normal`), and three fine-tuning tasks for benchmark datasets: object detection for MS-COCO, semantic segmentation for ADE20K, and single-label classification (`sinclass`) for ImageNet. The y-axis indicates the test error for each fine-tuning task. Dots indicate empirical results and dashed lines indicate the fitted curves of scaling law (1). For more details, see Section 4.2.

In this study, we find that the generalization error on fine-tuning is explained by a simple scaling law,

$$\text{test error} \simeq Dn^{-\alpha} + C, \tag{1}$$

where coefficient $D > 0$ and exponent $\alpha > 0$ describe the convergence speed of pre-training, and constant $C \geq 0$ determines the lower limit of the error. We refer to $\alpha$ as *pre-training rate* and $C$ as *transfer gap*. We can predict how large the pre-training data should be to achieve the desired accuracy by estimating the parameters $\alpha, C$ from the empirical results. Additionally, we analyze the dynamics of transfer learning using the recent theoretical results based on the neural tangent kernel (Nitanda & Suzuki, 2021) and confirm that the above law agrees with the theoretical analysis. We empirically validated our scaling law on various experimental settings of benchmark tasks, model sizes, and complexities of synthetic images.

Our contributions are summarized as follows.

- From empirical results and theoretical analysis, we elicit a law that describes how generalization scales in terms of data sizes on pre-training and fine-tuning.
- We confirm that the derived law explains the empirical results for various settings in terms of pre-training/fine-tuning tasks, model size, and data complexity (e.g., Figure 1). Furthermore, we demonstrate that we can use the estimated parameters in our scaling law to assess how much improvement we can expect from the pre-training procedure based on synthetic data.
- We theoretically derive a generalization bound for a general transfer learning setting and confirm its agreement with our empirical findings.

## 2 RELATED WORK

**Supervised pre-training for visual tasks**    Many empirical studies show that the performance at a fine-tuning task scales with pre-training data (and model) size. For example, Huh et al. (2016) studied the scaling behavior on ImageNet pre-trained models. Beyond ImageNet, Sun et al. (2017) studied the effect of pre-training with pseudo-labeled large-scale data and found a logarithmic scaling behavior. Similar results were observed by Kolesnikov et al. (2019).

**Syn2real transfer**    The utility of synthetic images as supervised data for computer vision tasks has been continuously studied by many researchers (Su et al., 2015; Movshovitz-Attias et al., 2016; Georgakis et al., 2017; Tremblay et al., 2018; Hinterstoisser et al., 2019; Borrego et al., 2018; Chen et al., 2021; Newell & Deng, 2020; Devaranjan et al., 2020; Mousavi et al., 2020; Hodaň et al., 2019). These studies found positive evidence that using synthetic images is helpful to the fine-tuning task. In addition, they demonstrated how data complexity, induced by e.g., light randomization, affects the final performance. For example, Newell & Deng (2020) investigated how the recent self-supervised methods perform well as a pre-training task to improve the performance of downstream tasks. In this paper, following this line of research, we quantify the effects under the lens of the scaling law (1).

**Neural scaling laws**    The scaling behavior of generalization error, including some theoretical works (e.g., Amari et al., 1992), has been studied extensively. For modern neural networks, Hestness et al. (2017) empirically observed the power-law behavior of generalization for language, image, and speech domains with respect to the training size. Rosenfeld et al. (2019) constructed a predictive form for the power-law in terms of data and model sizes. Kaplan et al. (2020) pushed forward this direction in the language domain, describing that the generalization of transformers obeys the power law in terms of a compute budget in addition to data and model sizes. Since then, similar scaling laws have been discovered in other data domains (Henighan et al., 2020). Several authors have also attempted theoretical analysis. Hutter (2021) analyzed a simple class of models that exhibits a power-law $n^{-\beta}$ in terms of data size $n$ with arbitrary $\beta > 0$. Bahri et al. (2021) addressed power laws under four regimes for model and data size. Note that these theoretical studies, unlike ours, are concerned with scaling laws in a non-transfer setting.

Hernandez et al. (2021) studied the scaling laws for general transfer learning, which is the most relevant to this study. A key difference is that they focused on fine-tuning data size as a scaling factor, while we focus on pre-training data size. Further, they found scaling laws in terms of the transferred effective data, which is converted data amount necessary to achieve the same performance gain by pre-training. In contrast, Eq. (1) explains the test error with respect to the pre-training data size directly at a fine-tuning task. Other differences include task domains (language vs. vision) and architectures (transformer vs. CNN).

**Theory of transfer learning**    Theoretical analysis of transfer learning has been dated back to decades ago (Baxter, 2000) and has been pursued extensively. Among others, some recent studies (Maurer et al., 2016; Du et al., 2020; Tripuraneni et al., 2020) derived an error bound of a fine-tuning task in the multi-task scenario based on complexity analysis; the bound takes an additive form $O(An^{-1/2} + Bs^{-1/2})$, where $n$ and $s$ are the data size of pre-training and fine-tuning, respectively, with coefficients $A$ and $B$. Neural network regression has been also discussed with this bound (Tripuraneni et al., 2020). In the field of domain adaptation, error bounds have been derived in relation to the mismatch between source and target input distributions (Ganin et al., 2016; Acuna et al., 2021). They also proposed algorithms to adopt a new data domain. However, unlike in this study, no specific learning dynamics has been taken into account. In the area of hypothesis transfer learning (Fei-Fei et al., 2006; Yang et al., 2007), among many theoretical works, Du et al. (2017) has derived a risk bound for kernel ridge regression with transfer realized as the weights on the training samples. The obtained bound takes a similar form to our scaling law. However, the learning dynamics of neural networks initialized with a pre-trained model has never been explored in this context.

## 3    SCALING LAWS FOR PRE-TRAINING AND FINE-TUNING

The main obstacle in analyzing the test error is that we have to consider interplay between the effects of pre-training and fine-tuning. Let $L(n, s) \geq 0$ be the test error of a fine-tuning task with pre-training data size $n$ and fine-tuning data size $s$. As the simplest case, consider a fine-tuning task without pre-training ($n = 0$), which boils the transfer learning down to a standard learning setting. In this case, the prior studies of both classical learning theory and neural scaling laws tell us that the test error decreases polynomially[1] with the fine-tuning data size $s$, that is, $L(0, s) = Bs^{-\beta} + \mathcal{E}$ with decay rate $\beta > 0$ and irreducible loss $\mathcal{E} \geq 0$. The irreducible loss $\mathcal{E}$ is the inevitable error given by

---

[1]For classification with strong low-noise condition, it is known that the decay rate can be exponential (Nitanda & Suzuki, 2019). However, we focus only on the polynomial decay without such strong condition in this paper.

the best possible mapping; it is caused by noise in continuous outputs or labels. Hereafter we assume $\mathcal{E} = 0$ for brevity.

## 3.1 INDUCTION OF SCALING LAW WITH SMALL EMPIRICAL RESULTS

To speculate a scaling law, we conducted preliminary experiments.[2] We pre-trained ResNet-50 by a synthetic classification task and fine-tuned by ImageNet. Figure 2 (a) presents the log-log plot of error curves with respect to pre-training data size $n$, where each shape and color indicates a different fine-tuning size $s$. It shows that the pre-training effect diminishes for large $n$. In contrast, Figure 2 (b) presents the relations between the error and the fine-tuning size $s$ with different $n$. It indicates the error

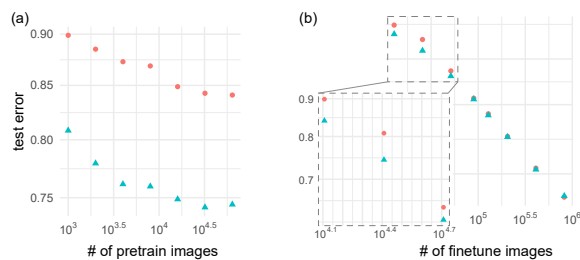

Figure 2: Scaling curves with different (a) pre-training size and (b) fine-tuning size.

drops straight down regardless of $n$, confirming the power-law scaling with respect to $s$. The above observations and the fact that $L(0, s)$ decays polynomially are summarized as follows.

**Requirement 1.** $\lim_{s \to \infty} L(n, s) = 0$.

**Requirement 2.** $\lim_{n \to \infty} L(n, s) = \text{const}$.

**Requirement 3.** $L(0, s) = Bs^{-\beta}$.

Requirements 1 and 3 suggest the dependency of $n$ is embedded in the coefficient $B = g(n)$, i.e., the pre-training and fine-tuning effects interact multiplicatively. To satisfy Requirement 2, a reasonable choice for the pre-training effect is $g(n) = n^{-\alpha} + \gamma$; the error decays polynomially with respect to $n$ but has a plateau at $\gamma$. By combining these, we obtain

$$L(n, s) = \delta(\gamma + n^{-\alpha})s^{-\beta}, \tag{2}$$

where $\alpha, \beta > 0$ are decay rates for pre-training and fine-tuning, respectively, $\gamma \geq 0$ is a constant, and $\delta > 0$ is a coefficient. The exponent $\beta$ determines the convergence rate with respect to fine-tuning data size. From this viewpoint, $\delta(\gamma + n^{-\alpha})$ is the coefficient factor to the power law. The influence of the pre-training appears in this coefficient, where the constant term $\delta\gamma$ comes from the irreducible loss of the pre-training task and $n^{-\alpha}$ expresses the effect of pre-training data size. The theoretical consideration in Section E.5 suggests that the rates $\alpha$ and $\beta$ can depend on both the target functions of pre-training and fine-tuning as well as the learning rate.

## 3.2 THEORETICAL DEDUCTION OF SCALING LAW

Next, we analyze the fine-tuning error from a purely theoretical point of view. To incorporate the effect of pre-training that is given as an initialization, we need to analyze the test error during the training with a given learning algorithm such as SGD. We apply the recent development by Nitanda & Suzuki (2021) to transfer learning. The study successfully analyzes the generalization of neural networks in the dynamics of learning, showing it achieves minmax optimum rate. The analysis uses the framework of the reproducing kernel Hilbert space given by the neural tangent kernel (Jacot et al., 2018).

For theoretical analysis of transfer, it is important to formulate a task similarity between pre-training and fine-tuning. If the tasks were totally irrelevant (e.g., learning MNIST to forecast tomorrow's weather), pre-training would have no benefit. Following Nitanda & Suzuki (2021), for simplicity of analysis, we discuss only a regression problem with square loss. We assume that a vector input $x$ and scalar output $y$ follow $y = \phi_0(x)$ for pre-training and $y = \phi_0(x) + \phi_1(x)$ for fine-tuning, where we omit the output noise for brevity; the task types are identical sharing the same input-output form, and task similarity is controlled by $\phi_1$.

We analyze the situation where the effect of pre-training remains in the fine-tuning even for large data size ($s \to \infty$). More specifically, the theoretical analysis assumes a regularization term as the

---

[2]The results are replicated from Appendix C.2; see the subsection for more details.

$\ell_2$-distance between the weights and the initial values, and a smaller learning rate than constant in the fine-tuning. Hence we control how the pre-training effect is preserved through the regularization and learning rate. Other assumptions made for theoretical analysis concern the model and learning algorithm; a two-layer neural network having $M$ hidden units with continuous nonlinear activation[3] is adopted; for optimization, the averaged SGD (Polyak & Juditsky, 1992), an online algorithm, is used for a technical reason.

The following is an informal statement of the theoretical result. See Appendix E for details. We emphasize that our result holds not only for syn2real transfer but also for transfer learning in general.

**Theorem 1** (Informal). *Let $\hat{f}_{n,s}(x)$ be a model of width $M$ pre-trained by $n$ samples $(x_1, y_1), \ldots, (x_n, y_n)$ and fine-tuned by $s$ samples $(x'_1, y'_1), \ldots, (x'_s, y'_s)$ where inputs $x, x' \sim p(x)$ are i.i.d. with the input distribution $p(x)$ and $y = \phi_0(x)$ and $y' = \varphi(x') = \phi_0(x') + \phi_1(x')$. Then the generalization error of the squared loss $L(n, s) = |\hat{f}_{n,s}(x) - \varphi(x)|^2$ is bounded from above with high probability as*

$$E_x L(n, s) \leq A_1 (c_M + A_0 n^{-\alpha}) s^{-\beta} + \varepsilon_M. \tag{3}$$

*$\varepsilon_M$ and $c_M$ can be arbitrary small for large $M$; $A_0$ and $A_1$ are constants; the exponents $\alpha$ and $\beta$ depend on $\phi_0$, $\phi_1$, $p(x)$, and the learning rate of fine-tuning.*

The above bound (3) shows the correspondence with the empirical derivation of the full scaling law (2). Note that the approximation error $\varepsilon_M$ is omitted in (2).

We note that the derived bound takes a multiplicative form in terms of the pre-training and fine-tuning effects, which contrasts with the additive bounds such as $A n^{-1/2} + B s^{-1/2}$ (Tripuraneni et al., 2020). The existing studies consider the situation where a part of a network (e.g., backbone) is frozen during fine-tuning. Therefore, the error of pre-training is completely preserved after fine-tuning, and both errors appear in an additive way. This means that the effect of pre-training is irreducible by the effect of fine-tuning, and vice versa. In contrast, our analysis deals with the case of re-optimizing the entire network in fine-tuning. In that case, the pre-trained model is used as initial values. As a result, even if the error in pre-training is large, the final error can be reduced to zero by increasing the amount of fine-tuning data.

### 3.3 INSIGHTS AND PRACTICAL VALUES

The form of the full scaling law (2) suggests that there are two scenarios depending on whether fine-tuning data is big or small. In "big fine-tune" regime, pre-training contributes relatively little. By taking logarithm, we can separate the full scaling law (2) into the pre-training part $u(n) = \log(n^{-\alpha} + \gamma)$ and the fine-tuning part $v(s) = -\beta \log s$. Consider to increase $n$ by squaring it. Since the pre-training part cannot be reduced below $\log(\gamma)$ as $u(n) > u(n^2) > \log(\gamma)$, the relative improvement $(u(n^2) - u(n))/v(s)$ becomes infinitesimal for large $s$. Figure 2 (b) confirms this situation. Indeed, prior studies provide the same conclusion that the gain from pre-training can easily vanish (He et al., 2018; Newell & Deng, 2020) or a target task accuracy even degrade (Zoph et al., 2020) if we have large enough fine-tuning data.

The above observation, however, does not mean pre-training is futile. Dense prediction tasks such as depth estimation require pixel-level annotations, which critically limits the number of labeled data. Pre-training is indispensable in such "small fine-tune" regime. Based on this, we hereafter analyze the case where the fine-tuning size $s$ is fixed. By eliminating $s$-dependent terms in (2), we obtain a simplified law (1) by setting $D = \delta s^{-\beta}$ and $C = \delta \gamma s^{-\beta}$. After several evaluations, these parameters including $\alpha$ can be estimated by the nonlinear least squares method (see also Section 4.1).

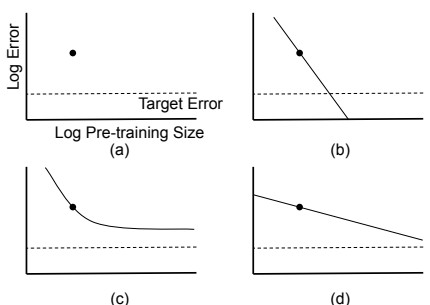

Figure 3: Pre-training scenarios.

As a practical benefit, the estimated parameters of the simplified law (1) bring a way to assess syn2real transfer. Suppose we want to solve a classification task that requires at least $90\%$ accuracy with limited labels.

---

[3]ReLU is not included in this class, but we can generalize this condition; see (Nitanda & Suzuki, 2021).

We generate some number of synthetic images and pre-train with them, and we obtain 70% accuracy as Figure 3 (a). How can we achieve the required accuracy? It depends on the parameters of the scaling law. The best scenario is (b) — transfer gap $C$ is low and pre-training rate $\alpha$ is high. In this case, increasing synthetic images eventually leads the required accuracy. In contrast, when transfer gap $C$ is larger than the required accuracy (c), increasing synthetic images does not help to solve the problem. Similarly, for low pre-training rate $\alpha$ (d), we may have to generate tremendous amount of synthetic images that are computationally infeasible. In the last two cases, we have to change the rendering settings such as 3D models and light conditions to improve $C$ and/or $\alpha$, rather than increasing the data size. The estimation of $\alpha$ and $C$ requires to compute multiple fine-tuning processes. However, the estimated parameters tell us whether we should increase data or change the data generation process, which can reduce the total number of trials and errors.

## 4 EXPERIMENTS

### 4.1 SETTINGS

For experiments, we employed the following transfer learning protocol. First, we pre-train a model that consists of backbone and head networks from random initialization until convergence, and we select the best model in terms of the validation error of the pre-training task. Then, we extract the backbone and add a new head to fine-tune all the model parameters. For notations, the task names of object detection, semantic segmentation, multi-label classification, single-label classification, and surface normal estimation are abbreviated as `objdet`, `semseg`, `mulclass`, `sinclass`, and `normal`, respectively. The settings for transfer learning are denoted by arrows. For example, `objdet→semseg` indicates that a model is pre-trained by object detection, and fine-tuned by semantic segmentation. The experiments were conducted on an in-house cluster containing NVIDIA V100 GPUs. The total amount of computation was approximately 1700 GPU days (200 for image rendering, 1300 for pre-training, and 200 for fine-tuning). All the results including Figure 1 are shown as log-log plots.

**Pre-training:** We prepared four tasks: `mulclass`, `objdet`, `semseg`, and `normal`. We used ResNet-based models, where backbones were ResNet-50, unless otherwise specified, and the head networks were customized for each task. Synthetic images for pre-training were generated by BlenderProc (Denninger et al., 2019), an image renderer that can handle several domain randomization methods. For rendering, we used the setting of the BOP challenge 2020 (Hodaň et al., 2020) as our default setting. We used 172 3D models, where ten objects appeared on average for each image. We applied texture randomization for walls and a floor, randomization for area and point lights, and randomization for the camera. In most cases, the models were pre-trained with 64,000 images. We trained all models for the same fixed number of iterations depending on pre-training tasks and selected the best models for fine-tuning, which were validated by another 1000 synthetic images generated in the same way.

**Fine-tuning:** We evaluated `sinclass` by ImageNet (Russakovsky et al., 2015), `objdet` by MS-COCO (Lin et al., 2014), and `semseg` by ADE20K (Zhou et al., 2016). The number of images used was 1% of each data set (roughly, 12,000 for ImageNet, 1000 for COCO, and 200 for ADE20K). We fine-tuned the pre-trained models with these subsets of data for a fixed number of iterations and reported the error metrics for validation sets at the last iteration. The metrics were top-1 accuracy for classification, mean mAP for MS-COCO, and mean IoU for ADE20K. These metrics take their values from 0 to 1, and we converted them into errors such as 1 - accuracy.[4]

**Curve fitting:** After obtaining the empirical errors $\hat{L}$, we estimated the parameters of (1) by non-linear least squares in the log-log space. We solved the minimization problem of $\sum_i |\log \hat{L}(n_i, s) - \log(D n_i^{-\alpha} + C)|^2$ with a fixed fine-tuning data size $s$ and pre-training data sizes $n_i = 2^i \times 1000$ for data point index $i = 0, \ldots, 6$. In the experiments, we empirically encountered some instability between $D$ and $\alpha$. We fixed $D = 0.48$ by the median values of $D$'s for all the settings and estimated $\alpha$ and $C$ independently for each case. We explain this procedure with more details in Appendix D.

---

[4] Although the cross-entropy loss is commonly used, several studies (Sharma & Kaplan, 2020; Bahri et al., 2021) show that the scaling laws also hold for 1 - accuracy.

### 4.2 SCALING LAW UNIVERSALLY EXPLAINS DOWNSTREAM PERFORMANCE FOR VARIOUS TASK COMBINATIONS

Figure 1 shows the test errors of each fine-tuning task and fitted learning curves with Eq. (1), which describes the effect of pre-training data size $n$ for all combinations of pre-training and fine-tuning tasks. The scaling law fits with the empirical fine-tuning test errors with high accuracy in most cases.

### 4.3 BIGGER MODELS REDUCE THE TRANSFER GAP

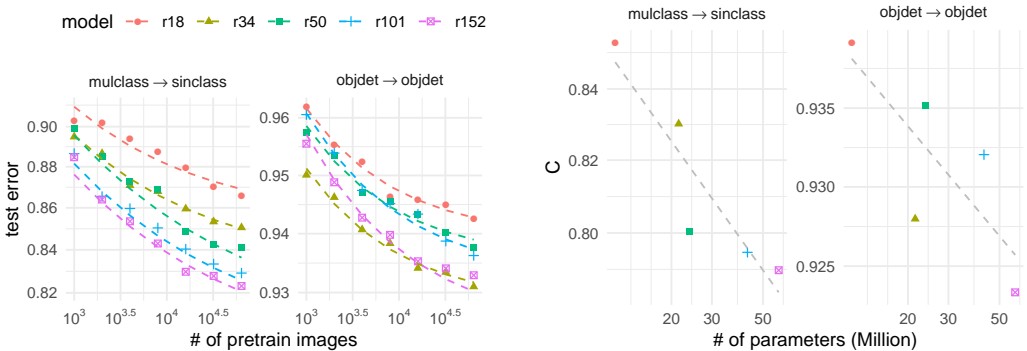

Figure 4: Effect of model size. Best viewed in color. **Left**: The scaling curves for mulclass→sinclass and objdet→objdet cases. The meanings of dots and lines are the same as those in Figure 1. **Right**: The estimated transfer gap $C$ (y-axis) versus the model size (x-axis) in log-log scale. The dots are estimated values, and the lines are linear fittings of them.

We compared several ResNet models as backbones in mulclass→sinclass and objdet→objdet to observe the effects of model size. Figure 4 (left) shows the curves of scaling laws for the pre-training data size $n$ for different sizes of backbone ResNet-$x$, where $x \in \{18, 34, 50, 101, 152\}$. The bigger models attain smaller test errors. Figure 4 (right) shows the values of the estimated transfer gap $C$. The results suggest that there is a roughly power-law relationship between the transfer gap and model size. This agrees with the scaling law with respect to the model size shown by Hernandez et al. (2021).

### 4.4 SCALING LAW CAN EXTRAPOLATE FOR MORE PRE-TRAINING IMAGES

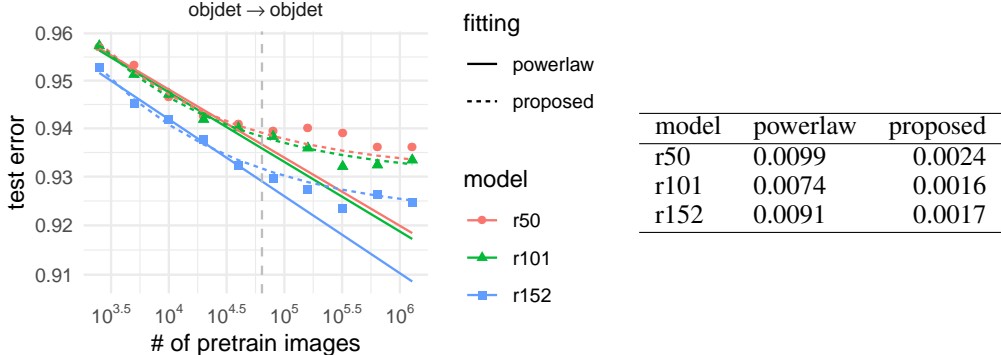

| model | powerlaw | proposed |
|-------|----------|----------|
| r50   | 0.0099   | 0.0024   |
| r101  | 0.0074   | 0.0016   |
| r152  | 0.0091   | 0.0017   |

Figure 5: Ability to extrapolate. **Left:** The solid lines represent the fitted power law and the dashed curves represent the fitted scaling law (1), in which the laws were fitted using the empirical errors where the pre-training size $n$ was less than $64,000$ (the first five dots). The vertical dashed line indicates where $n = 64,000$. **Right:** The root-mean-square errors between the laws and the actual test errors in the area of extrapolation (the last five dots).

We also evaluated the extrapolation ability of the scaling law. We increased the number of synthetic images from the original size ($n = 64{,}000$) to 1.28 million, and see how the fitted scaling law predicts the unseen test errors where $n > 64{,}000$. As a baseline, we compared the power-law model, which is equivalent to the derived scaling law (1) with $C = 0$. Figure 5 (left) shows the extrapolation results in `objdet`→`objdet` setting, which indicates the scaling law follows the saturating trend in regions with large pre-training sizes for all models, while the power-law model fails to capture it. The prediction errors is numerically shown in Figure 5 (right), which again shows our scaling law achieves better prediction performance.

## 4.5 DATA COMPLEXITY AFFECTS BOTH PRE-TRAINING RATE AND TRANSFER GAP

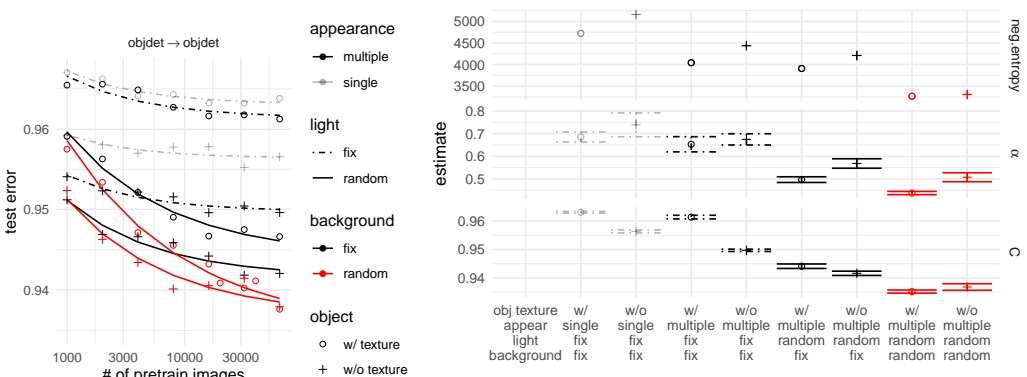

Figure 6: Effect of synthetic image complexity. Best viewed in color. **Left**: Scaling curves of different data complexities. **Right**: Estimated parameters. The error bars represent the standard error of the estimate in least squares.

We examined how the complexity of synthetic images affects fine-tuning performance. We controlled the following four rendering parameters: *Appearance*: Number of objects in each image; `single` or `multiple` (max 10 objects). *Light*: Either an area and point light is `randomized` or `fixed` in terms of height, color, and intensity. *Background*: Either the textures of floor/wall are `randomized` or `fixed`. *Object texture*: Either the 3D objects used for rendering contain texture (`w/`) or not (`w/o`). Indeed, the data complexity satisfies the following ordered relationships: `single` < `multiple` in *appearance*, `fix` < `random` in *light* and *background*, and `w/o` < `w/` in *object texture*[5]. To quantify the complexity, we computed the negative entropy of the Gaussian distribution fitted to the last activation values of the backbone network. For this purpose, we pre-trained ResNet-50 as a backbone with MS-COCO for 48 epochs and computed the empirical covariance of the last activations for all the synthetic data sets.

The estimated parameters are shown in Figure 6, which indicates the following (we discuss the implications of these results further in Section 5.1).

- Data complexity controlled by the rendering settings correlates with the negative entropy, implying the negative entropy expresses the actual complexity of pre-training data.
- Pre-training rate $\alpha$ correlates with data complexity. The larger complexity causes slower rates of convergence with respect to the pre-training data size.
- Transfer gap $C$ mostly correlates negatively with data complexity, but not for *object texture*.

As discussed in Section 4.1, we have fixed the value of $D$ to avoid numerical instability, which might cause some bias to the estimates of $\alpha$. We postulate, however, the value of $D$ depends mainly on the fine-tuning task and thus has a fixed value for different pre-training data complexities. This can be inferred from the theoretical analysis in Appendix E.5: the exponent $\beta$ in the main factor $s^{-\beta}$ of $D$ does not depend on the pre-training data distribution but only on the fine-tuning task or the pre-training true mapping. Thus, the values of $D$ should be similar over the different complexities, and the correlation of $\alpha$ preserves.

---

[5]The object category of `w/o` is a subset of `w/`, and `w/` has a strictly higher complexity than `w/o`.

## 5 Conclusion and Discussion

In this paper, we studied how the performance on syn2real transfer depends on pre-training and fine-tuning data sizes. Based on the experimental results, we found a scaling law (1) and its generalization (2) that explain the scaling behavior in various settings in terms of pre-training/fine-tuning tasks, model sizes, and data complexities. Further, we present the theoretical error bound for transfer learning and found our theoretical bound has a good agreement with the scaling law.

### 5.1 Implication of complexity results in Section 4.5

The results of Section 4.5 has two implications. First, data complexity (i.e., the diversity of images) largely affects the pre-training rate $\alpha$. This is reasonable because if we want a network to recognize more diverse images, we need to train it with more examples. Indeed, prior studies (Sharma & Kaplan, 2020; Bahri et al., 2021) observed that $\alpha$ is inversely proportional to the intrinsic dimension of the data (e.g., dimension of the data manifold), which is an equivalent concept of data complexity.

Second, the estimated values of the transfer gap $C$ suggest that increasing the complexity of data is generally beneficial to decrease $C$, but not always. Figure 6 (right) shows that increasing complexities in terms of *appearance*, *light*, and *background* reduces the transfer gap, which implies that these rendering operations are most effective to cover the fine-tuning task that uses real images. However, the additional complexity in *object texture* works negatively. We suspect that this occurred because of *shortcut learning* (Geirhos et al., 2020). Namely, adding textures to objects makes the recognition problem falsely easier because we can identify objects by textures rather than shapes. Because CNNs prefer to recognize objects by textures (Geirhos et al., 2018; Hermann et al., 2019), the pre-trained models may overfit to learn the texture features. Without object textures, pre-trained models have to learn the shape features because there is no other clue to distinguish the objects, and the learned features will be useful for real tasks.

### 5.2 Lessons to transfer learning and synthetic-to-real generalization

Our results suggest the transfer gap $C$ is the most crucial factor for successful transfer learning because $C$ determines the maximum utility of pre-training. Large-scale pre-training data can be useless when $C$ is large. In contrast, if $C$ is negligibly small, the law is reduced essentially to $n^{-\alpha}$, which tells that the volume of pre-training data is directly exchanged to the performance of fine-tuning tasks. Our empirical results suggest two strategies for reducing $C$: 1) Use bigger models and 2) fill the domain gap in terms of the decision rule and image distribution. For the latter, existing techniques such as domain randomization (Tobin et al., 2017) would be helpful.

### 5.3 Limitations of this study

- In the experiments, the scale of data is relatively limited (million-scale, not billion).
- We only examined ResNet as a network architecture (no Transformers).
- Although there are various visual tasks, our study only covers a few of them. Extending our observations to other visual tasks such as depth estimation, instance segmentation, and keypoint detection, as well as to other data domains such as language is future work.
- The theoretical results assume several conditions that may contradict the actual setting in the experiments. For example, our theory relies on ASGD instead of vanilla SGD. Also, the task types are assumed to be identical for the pre-training and fine-tuning tasks.
- In this study, we focus on finding a general rule of transfer learning, rather than improving absolute performance on specific tasks. We used popular vision tasks such as classification and ready-made rendering settings that is not designed to pre-train for the tasks. We expect to observe more performance gain with other syn2real-friendly tasks such as optical flow and elaborate rendering settings in future work.

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

# Appendix

## A    TRAINING DETAILS

### A.1    OBJECT DETECTION

We used Faster-RCNN (Ren et al., 2016) with FPN (Lin et al., 2017) as object detection models and ResNet (Goyal et al., 2017) as a backbone network of Faster-RCNN.

We used the following training procedure: We trained the model using momentum SGD of momentum $0.9$ with weight decay of $10^{-4}$. The global batch size was set to $64$ when training ResNet18, ResNet34, ResNet50, and ResNet101. The batch size was set to $32$ when training ResNet152 to avoid out-of-memory errors. The batch statistics in batch normalization layers were computed across all GPUs. We used a base image size of $640 \times 640$ in the same way as YOLACT training (Bolya et al., 2019). We used mixed16 training to reduce the memory footprint. We also adopted random horizontal flipping as data augmentation to images. The learning rate was set to $0.02$, and we used the cosine decay with a warmup scheme. The warmup length is $120{,}000$ images ($3{,}750$ iterations for ResNet152 and $1{,}875$ iterations for other models). As for evaluation, we followed the standard settings in COCO dataset (Ren et al., 2016).

We pre-trained the model with $14{,}400{,}000$ images ($450{,}000$ iterations for ResNet152 and $225{,}000$ iterations for other models). We used the models that achieved the best mmAP as the initial value of fine-tuning.

We used COCO (Lin et al., 2014) as the fine-tuning dataset. We trained the model with $1{,}440{,}000$ images ($45{,}000$ iterations for ResNet152 and $22{,}500$ iterations for other models) during fine-tuning.

### A.2    SEMANTIC SEGMENTATION

We used DeepLabV3 (Chen et al., 2017) with the softmax cross-entropy loss as the semantic segmentation model and ResNet50 (Goyal et al., 2017) as its backbone. The model configuration follows the implementation in torchvision[6]. It should be noted that DeepLabV3 requires *dilated* ResNet as the backbone, which is not the case in object detection and classification tasks. Even though, the shapes of weight tensors of dilated ResNet50 exactly match those of non-dilated ResNet50; thus we can use the pre-trained weights of dilated and non-dilated ResNet50 interchangeably.

The learning procedure is based on the reference implementation[7] of torchvision. We added an auxiliary branch based on FCN (Long et al., 2015) which takes `conv4` of the backbone as the input. In the computation of loss function, the loss for the auxiliary branch is computed in the same way as for the main branch and is added to the overall loss after multiplying by the factor $0.5$. The model was trained using momentum SGD of momentum $0.9$ with weight decay of $10^{-4}$. The global batch size was set to $32$. The batch statistics in batch normalization layers were computed across all GPUs. During training, images were first resized so that the length of the shorter edge becomes an integer uniformly chosen from $[520 \times 0.5, 520 \times 2]$, then horizontally flipped with probability $0.5$, finally randomly cropped to $480 \times 480$. The learning rate (LR) was decayed according to the polynomial LR schedule of rate $0.9$ and initial LR of $0.02$. For the parameters of the auxiliary classifier, the LR was multiplied by $10$. The evaluation was performed once every $3{,}125$ iterations (almost equivalent to $5$ epochs in full ADE20K). In the evaluation, images were resized so that the length of the shorter edge becomes $520$.

In pre-training, we trained $125{,}000$ iterations which roughly equals $200$ epochs in full ADE20K (Zhou et al., 2017). We used the model that achieved the best mIoU as the initial value of fine-tuning. We pre-trained models using our synthetic datasets. When training with them, backgrounds (points at which no foreground objects were present) were also considered to be a separate class in semantic segmentation.

---

[6]`https://github.com/pytorch/vision`
[7]`https://github.com/pytorch/vision/tree/master/references/segmentation`

We used the ADE20K (Zhou et al., 2017) datasets as the fine-tuning target. In fine-tuning, we trained the model for 18,750 iterations, which correspond to 30 epochs of full ADE20K. The metric was mIoU score.

### A.3 MULTI-LABEL CLASSIFICATION

We used ResNet (Goyal et al., 2017) with binary cross-entropy used as the loss function in multi-label classification. We used the following training procedure: We trained the model using momentum SGD of momentum $0.9$ with weight decay of $10^{-4}$. The batch size was set as 32 per GPU, thus 256 in total. We trained the models for 112,500 iterations. The input size of the images was simply resized to $640 \times 640$. We adopted random horizontal flipping as data augmentation to images. The learning rate was set to $0.1$. The cosine decay with a warmup scheme was used. The warmup length was 120,000 images. The evaluation was performed once every 120,000 images. In the evaluation, the image size was the same as that used during training, and data augmentation was not used. We used the mAP score as the metric.

### A.4 SINGLE-LABEL CLASSIFICATION

As in multi-label classification, we used ResNet. The softmax cross-entropy was used as the loss function of the single-label classification. The learning procedure is based on (Goyal et al., 2017). However, we used cosine decay for the learning rate scheduling.

### A.5 SURFACE NORMAL ESTIMATION

As in semantic segmentation, we used DeepLabV3 (Chen et al., 2017) as the model for surface normal estimation. The model configuration and the training procedure were exactly the same as in semantic segmentation, except for the following changes:

- Dimension of output channels was changed to 3, each of which corresponds to the 3 axes of the normal vector,
- Initial LR was changed to $0.04$,
- Length of pre-training was 200,000 iterations, which corresponds to $100$ epochs in our synthetic dataset,
- Random flipping was not performed during the data augmentation, and
- Loss function was the average of the value, $1.0 - \mathrm{n} \cdot \hat{\mathrm{n}}$, which was computed for each valid pixel where $\mathrm{n}$ is the ground-truth normal vector and $\hat{\mathrm{n}}$ is the model output (after L2 normalization).

## B SYNTHETIC DATA DETAILS

The data generation strategy is based on the "on surface sampling" setting in the BoP challenge dataset[8]. In this setting, the sampled objects will be spawned in a cube-shaped room with one point light and one surface light. As the objects to be spawn, we used all the BoP object sets, i.e., LM, T-LESS, ITODD, HB, YCB-V, RU-APC, IC-BIN, IC-MI, TUD-L, and TYO-L.[9] There are 173 objects in total. After generated a random scene (position of objects, lights, etc), we took 10 pictures by 10 different camera poses. This means that, if we have 10K images in total, there are 1K unique scenes, and 9K images are inflated by just changing the camera angle and position.

To control the data complexity, we selected four attributes in the generation strategy and prepared two options for each.

*Appearance* controls how many objects are generated in a room in a single scene. For each scene, we randomly select ten objects for the **multiple** setting and one object for the **single** setting.

*Light* controls the light sources. In the **random** setting, the color, height, and strength of lights are randomized. In contrast, in the **fix** setting, they are all fixed.

---

[8]https://github.com/DLR-RM/BlenderProc/tree/main/examples/bop_challenge
[9]https://bop.felk.cvut.cz/datasets/

*Background* controls the texture of the room, i.e., floor and walls. In the **random** setting, we assign a random PBR material from the CC0 Textures[10] library, and we selected one carpet texture for the **fix** setting.

*Object texture* controls the object set to be used. The BoP object set is consists of several types of object sets as described above. Among them, T-LESS and ITODD consist of industry-relevant objects and they do not have textures and colors.[11] For the **w/o** setting, we only used such texture-less objects to be sampled. In contrast, we sample all the 173 objects include T-LESS and ITODD in the **w/** setting.

We generated eight variations of datasets by changing these attributes, which were used in the experiments of Section 4.5. Figure 7 shows the example of generated images with the value of each attribute.

---

[10] https://ambientcg.com

[11] T-Less and ITODD contain 30 and 28 objects, respectively.

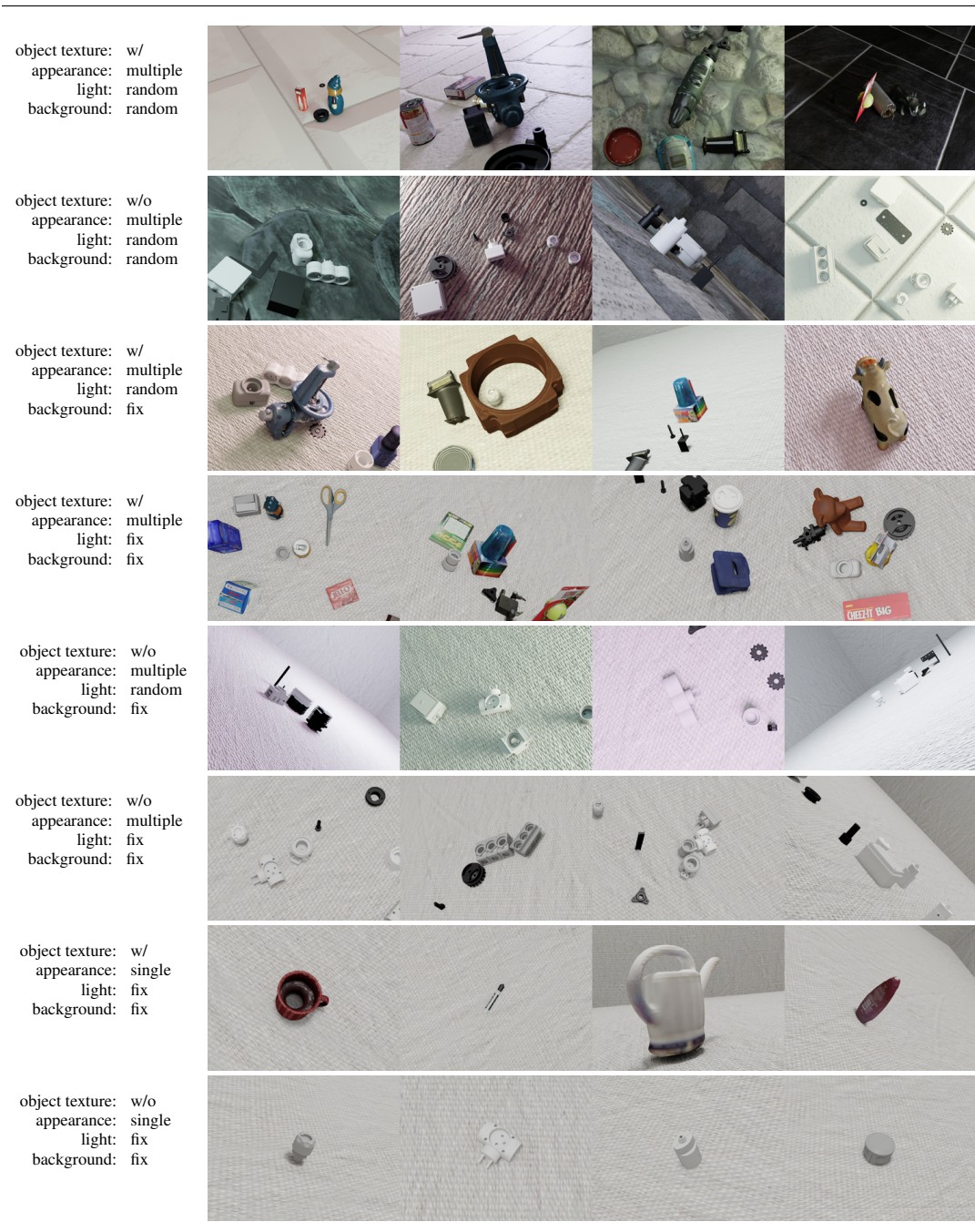

object texture: w/
appearance: multiple
light: random
background: random

object texture: w/o
appearance: multiple
light: random
background: random

object texture: w/
appearance: multiple
light: random
background: fix

object texture: w/
appearance: multiple
light: fix
background: fix

object texture: w/o
appearance: multiple
light: random
background: fix

object texture: w/o
appearance: multiple
light: fix
background: fix

object texture: w/
appearance: single
light: fix
background: fix

object texture: w/o
appearance: single
light: fix
background: fix

Figure 7: Example of generated datasets

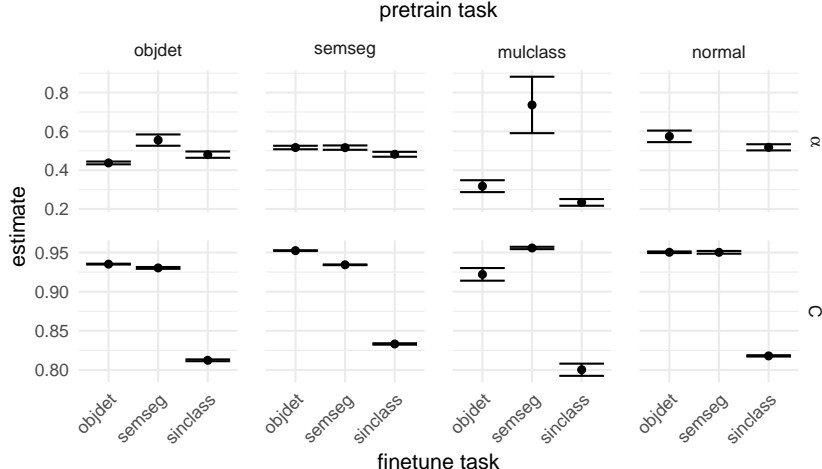

Figure 8: The estimated values of the pre-training rate $\alpha$ and the transfer gap $C$ in the cross-task setting (as the same as Figure 1). The error bars present the standard error of the estimates in the least squares.

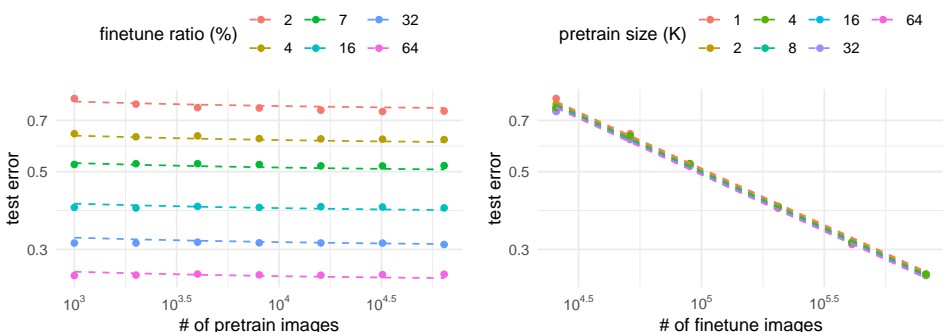

Figure 9: Empirical and fitting results for various pre-train and fine-tune data sizes in mulclass→sinclass. All curves are fitted using the full law (2). Best viewed in color. **Left**: Effect of pre-training data size (x-axis) for fixed fine-tuning data sizes. **Right**: Effect of fine-tuning data size (x-axis) for fixed pre-training data sizes.

## C ADDITIONAL EXPERIMENTS

### C.1 ESTIMATED PARAMETERS IN THE CROSS-TASK SETTING

Figure 8 shows the estimated parameters $(\alpha, C)$ at the experiments described in Section 4.2. Note that the result of $\alpha$ at normal→semseg is omitted because its estimated value is highly unstable (the standard deviation is larger than 1).

### C.2 FULL SCALING LAW COLLECTIVELY RELATES PRE-TRAINING AND FINE-TUNING DATA SIZE

Next, we verify the validity of the full scaling law (2). In the mulclass→sinclass setting with ResNet-50, we changed the fine-tuning data size from 2% to 64% of the ImageNet.[12] We then fitted all results by a single equation (2) to estimate the parameters except the irreducible loss $\mathcal{E}$

---

[12]ImageNet contains a class imbalance problem. If we use 100% of the ImageNet, we cannot provide the same sample size per class. To eliminate the effect of class imbalance, we made the sampling ratio to keep the balance up to 64% and excluded the case of 100%.

(we assumed $\mathcal{E} = 0$ from the preliminary results in Figure 2). The results in Figure 9 show that all empirical test errors are explained remarkably well by Eq. (2), which has only four parameters to fit in this case. The estimated parameters are $\alpha = 0.544$, $\beta = 0.322$, $\gamma = 0.478$, and $\delta = 41.8$.

## C.3 LINEARIZED RESULTS

The transfer gap $C$ in (1) causes a plateau of the scaling law. Conversely, if we subtract the estimated $C$ from the results, we must be able to recover the power-law scaling. To confirm this, we subtracted the estimated $C$ from the empirical errors $\hat{L}$ of the previous results. Figures 10–12 show the modified version of scaling law fittings. Overall, the empirical errors behave linearly along with the estimated power-law term $Dn^{-\alpha}$. Note that, in `mulclass`→`semseg` and `normal`→`semseg`, a few points of $\hat{L}$ become negative after subtracting $C$, and these points are not depicted.

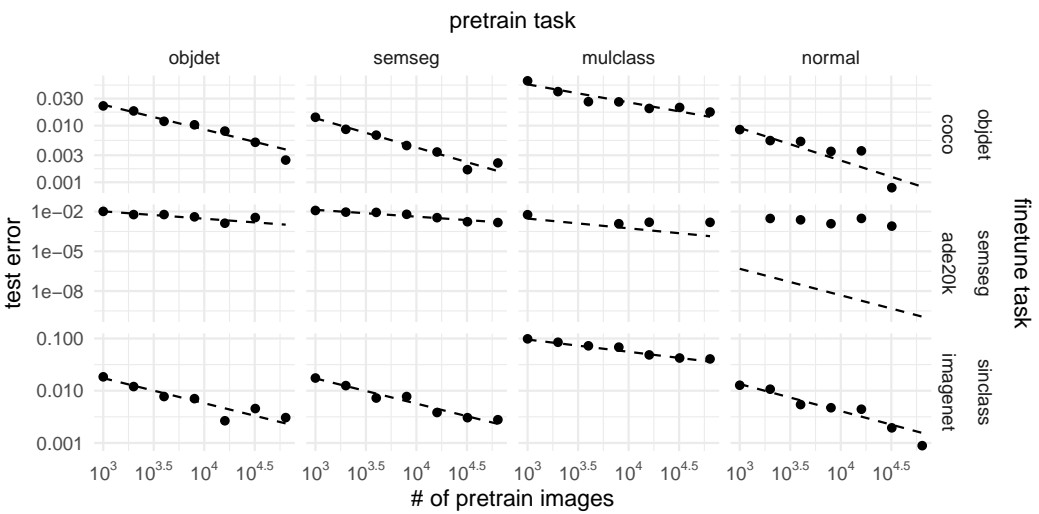

Figure 10: The linearized version of Figure 1.

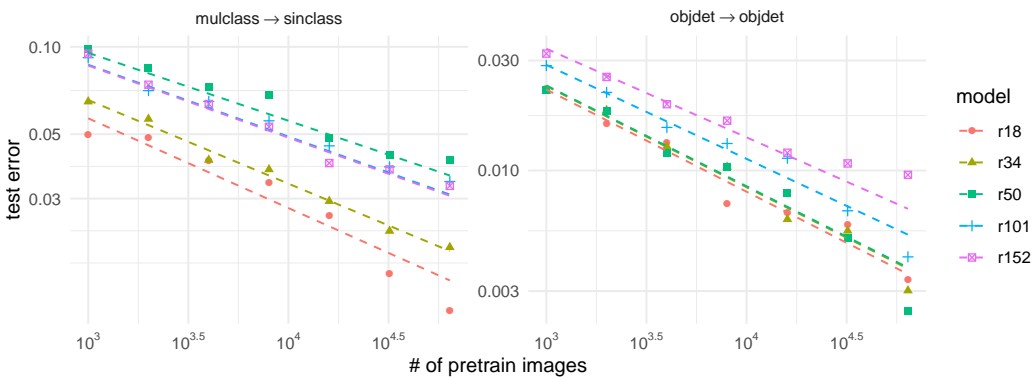

Figure 11: The linearized version of Figure 4.

## D  EMPIRICAL DEPENDENCY BETWEEN $\alpha$ AND $D$

When $C$ is non-zero, the joint estimation of $D$ and $\alpha$ in (1) have an issue of numerical stability due to the small number of observations and noise, which can cause high dependence on each other. Figure 13 (left) shows the curves of (1) with $C = 0.5$, where the solid red curve is $D = 0.5, \alpha = 0.4$ and the dashed blue line is $D = 1, \alpha = 0.5$. We see that both curves are almost indistinguishable for

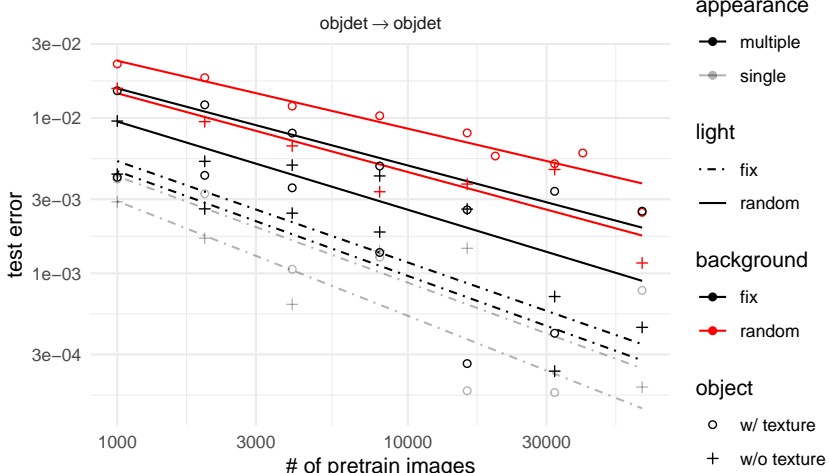

Figure 12: The linearized version of Figure 6.

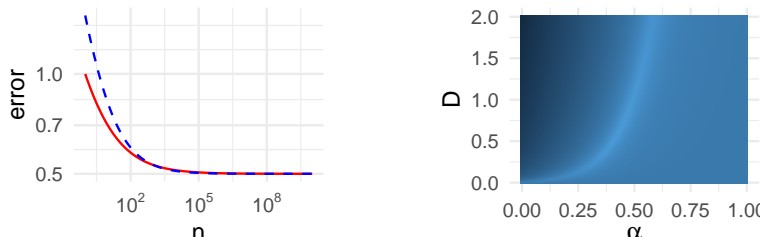

Figure 13: Examples of parameter dependency.

a large $n$. Figure 13 (right) shows the actual landscape in terms of $\alpha$ and $D$ of the nonlinear least-squares at `objdet→objdet`, the bright areas indicate the fitting loss is small. We see that there is a quadratic-like trajectory in the landscape, which implies the solutions are somehow redundant. Similar landscapes were observed for other tasks (Figure 14).

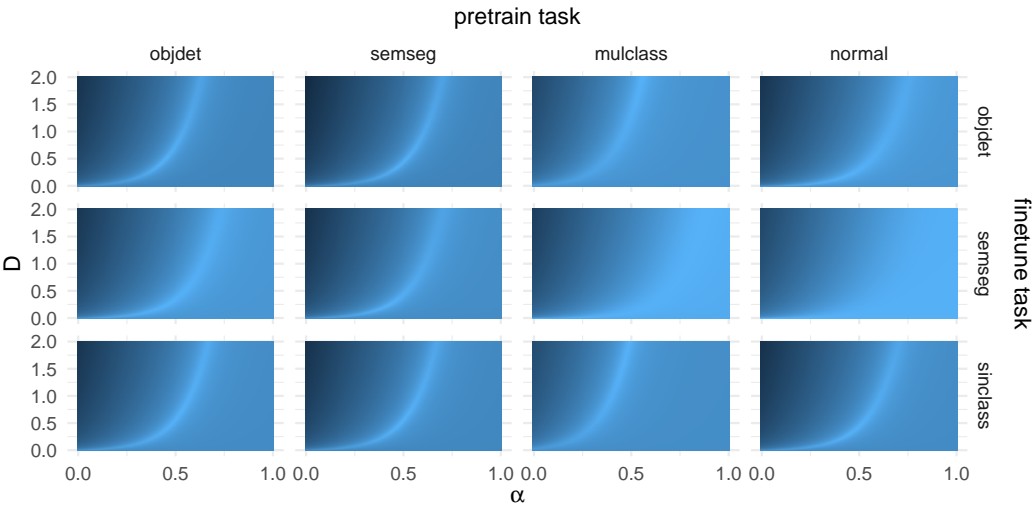

Figure 14: Loss landscapes of curve fittings.

To avoid this issue, we fixed a common $D$ for all the cases and estimated $\alpha$ for each. To determine $D$, we used the following procedure. First, we prepared two global parameters $\hat{\alpha}, \hat{D}$ and set $0.5$ as their initial values. Then, we fitted the curves by two equations, $Dn^{-\hat{\alpha}} + C$ and $\hat{D}n^{-\alpha} + C$, and estimated $\alpha$ and $D$. Next, we computed the median of $\alpha$ and substituted them into $\hat{\alpha}$. We did the same for $D$ and $\hat{D}$. After a few iterations, we got a converged value of $\hat{D} = 0.48$. In the experiments, we used the value for $D$ and fixed it.

# E    DETAILS OF THEORETICAL ANALYSIS

This section gives details of the theoretical discussions given in Section 3.2. For the analysis of learning and generalization bound, we use the techniques developed recently by Nitanda & Suzuki (2021). There are many works on the generalization of neural networks. To list a few, Neyshabur et al. (2015), Neyshabur et al. (2017), Bartlett et al. (2017), Wei & Ma (2020), and Suzuki (2018) analyze the generalization of neural networks based on complexity bounds. These generalization bounds, however, do not consider an algorithm of learning, such as stochastic gradient descent (SGD). Recently, learning dynamics of neural networks has been analyzed based on Neural Tangent Kernel (NTK) Jacot et al. (2018) and global convergence of wide neural networks has been revealed Allen-Zhu et al. (2018); Du et al. (2019). Based on the NTK framework, Arora et al. (2019) and Nitanda et al. (2020) showed a generalization bound of the gradient descent learning of neural networks. More recently, Nitanda & Suzuki (2021) focused the functional space given by NTK and showed that the two-layer neural network with averaged SGD achieves the minimax optimal rate with respect to the function class used in the standard theory of function estimation with kernels. We employ the method of Nitanda & Suzuki (2021), which is the most suitable for our analysis of transfer learning: it enables to examine the dependence on the initial parameter in the learning, and avoids the assumption of a positive margin of eigenvalues used in Arora et al. (2019) and Nitanda et al. (2020).

## E.1    PROBLEM SETTING

In the pre-training, the task is to learn the target function $\phi_0$ with $T_0$ training data $(\tilde{x}_i, \tilde{y}_i)_{i=1}^{T_0}$, where $y_i = \phi_0(x_i)$, while in the fine-tuning phase, the network is initialized by the final parameter learned by the pre-training, and the whole parameter is updated in the training. We assume that the target function in the fine-tuning is given by

$$\varphi(x) = \phi_0(x) + \phi_1(x), \tag{4}$$

and $T_1$ training data is given by $(x_j, y_j)_{j=1}^{T_1}$ with $y_j = \varphi(x_j)$. In this setting, the goal of the fine-tuning phase will be to learn the additional function $\phi_1$ mainly. Note that, for simplicity of analysis, we assume noiseless training data, i.e., we assume the supervised signal $y_j$ is given by a deterministic function of $x_j$, but extension to more general cases is not difficult as discussed in Nitanda & Suzuki (2021). In the analysis, the data are assumed to satisfy $x \in \mathbb{R}^d$, $\|x\|_2 = 1$ and $y \in [0, 1]$. The distribution of the input data $x$ is denoted by $\rho_X$, and the same for the pre-training and fine-tuning.

For tractable theoretical analysis, we consider a simple scalar-valued two-layer neural network model with $M$ hidden units:

$$g_\Theta(x) = \frac{1}{\sqrt{M}} \sum_{r=1}^{M} a_r \sigma(b_r^T x). \tag{5}$$

We omit the bias term, but with obvious modification, it is not difficult to include it (see Nitanda & Suzuki (2021)).

As in Nitanda & Suzuki (2021), we consider the averaged stochastic gradient descent (ASGD), where one training sample is given at every time step for the stochastic gradient descent as in online learning, and all the parameters in the time course are averaged after the final time step for the inference, that is, after proceeding up to prescribed $T$ ($T = T_0$ or $T_1$) time steps, the parameter to be used in the inference is given by

$$\overline{\Theta}^{(T)} := \frac{1}{T+1} \sum_{t=0}^{T} \Theta^{(t)}. \tag{6}$$

The final network uses this averaged parameter, i.e., the final network is given by $g_{\overline{\Theta}^{(T)}}(x)$.

The parameter is initialized as $\Theta^{(0)} = (a_1^{(0)}, b_1^{(0)}, \ldots, a_M^{(0)}, b_M^{(0)})$. For the pre-trainig, each $b^{(0)}$ is independently given by the uniform distribution on the unit sphere. As in Nitanda & Suzuki (2021), $a^{(0)}$ are initilized as 1 or $-1$ so that $g_{\Theta^{(0)}} = 0$. As explained before, the initial parameter of the fine-tuning is the same as the averaged parameter of the pre-training $\overline{\Theta}_{pre}^{(T_0)}$. The objective function to minimize for the pre-training and fine-tuning is given by the following regularized empirical risk:

$$L(\Theta) := \frac{1}{2} \sum_i (y_i - g_\Theta(x_i))^2 + \frac{\lambda}{2} \Big\{ \|a - a^{(0)}\|_2^2 + \sum_r \|b_r - b_r^{(0)}\|_2^2 \Big\}, \tag{7}$$

where $\lambda$ is the regularization coefficient, which is a hyperparameter. The values of $\lambda$ in the pre-training and fine-tuning can be different, and denoted by $\lambda_0$ and $\lambda_1$, respectively. Note that the regularization in Eq. (7) is not the most common $\ell_2$-regularization, where $\|\Theta\|_2^2$ is used for the regularization. When applied in fine-tuning, however, the above regularization can be interpreted as *elastic weight consolidation* (Kirkpatrick et al., 2017), which prevents forgetting the pre-trained parameters.

We consider online learning, in which at every step $t$ ($t \leq T - 1$), one datum $x_i$ is sampled from $\rho_X$ independently, and $(x_i, y_i)$ (or $(\tilde{x}_i, \tilde{y}_i)$ in pre-training) is used to update the parameter according to the gradient descent:

$$\Theta^{(t+1)} = \Theta^{(t)} - \eta \frac{\partial L(\Theta^{(t)})}{\partial \Theta}, \tag{8}$$

where $\eta$ is a learning rate. More explicitly,

$$a_r^{(t+1)} - a_r^{(0)} = (1 - \eta\lambda)(a_r^{(t)} - a_r^{(0)}) - \frac{\eta}{\sqrt{M}}(g_{\Theta^{(t)}} - y_t)\sigma(b_t^{(t)T} x_t),$$

$$b_r^{(t+1)} - b_r^{(0)} = (1 - \eta\lambda)(b_r^{(t)} - b_r^{(0)}) - \frac{\eta}{\sqrt{M}}(g_{\Theta^{(t)}} - y_t)a_r\sigma'(b_t^{(t)T} x_t)x_t. \tag{9}$$

### E.2 NEURAL TANGENT KERNEL

In the theoretical analysis, the neural tangent kernel (NTK, Jacot et al., 2018) is used for approximating the dynamics of ASGD by a linear functional recursion on the corresponding function space. The NTK of this model is given by

$$k_\infty(x, x') = E_{b^{(0)}}[\sigma(b^{(0)T}x)\sigma(b^{(0)T}x')] + x^T x' E_{b^{(0)}}[\sigma'(b^{(0)T}x)\sigma'(b^{(0)T}x')]. \tag{10}$$

The positive definite kernel $k_\infty$ naturally defines a reproducing kernel Hilbert space (RKHS), which is denoted by $\mathcal{H}_\infty$.

The integral operator $\Sigma_\infty$ on $L^2(\rho_X)$ is defined by

$$\Sigma_\infty f := \int k_\infty(\cdot, x) f(x) d\rho_X(x). \tag{11}$$

It is known that $\Sigma_\infty$ admits eigendecomposition

$$\Sigma_\infty \psi_s = \gamma_s \psi_s, \tag{12}$$

where $\psi_s$ is an eigenvector with $\|\psi_s\|_{L^2(\rho_X)} = 1$ and $\gamma_1 \geq \gamma_2 \geq \ldots > 0$ are eigenvalues in descending order. Mercer's theorem tells that $k_\infty$ has an expansion:

$$k_\infty(x, x') = \sum_{s=1}^\infty \gamma_s \psi_s(x) \psi_s(x'),$$

where the convergence is understood as in $L^2(\rho_X)$ for general, and absolutely and uniformly if $\rho_X$ is a uniform distribution on a compact set.

### E.3 ASSUMPTIONS

For theoretical analysis, we make the following assumptions. For an operator $\Sigma$, the range of $\Sigma$ is denoted by $\mathcal{R}(\Sigma)$.

(A1) The activation function $\sigma$ is differentiable up to the second order, and there exists $C > 0$ such that $\|\sigma''\|_\infty \leq C$, $\|\sigma'\|_\infty \leq 2$, and $|\sigma(u)| \leq 1 + |u|$ for $\forall u \in \mathbb{R}$.

(A2) $\text{supp}(\rho_X) \subset \{x \in \mathbb{R}^d \mid \|x\| \leq 1\}$ and $y \in [-1, 1]$.

(A3) There exist $1/2 \leq r_0, r_1 \leq 1$ such that $\phi_0 \in \mathcal{R}(\Sigma_\infty^{r_0})$ and $\phi_1 \in \mathcal{R}(\Sigma_\infty^{r_1})$.

(A4) There exists $\xi > 1$ such that $\gamma_\ell = \Theta(\ell^{-\xi})$.

As in Nitanda & Suzuki (2021), Assumption (A1) assumes that the activation $\sigma$ is differentiable in this paper. In Nitanda & Suzuki (2021), however, they have developed a theory on how to extend the results to the case of ReLU by approximating it with a smooth function. It is well known that Assumption (A4) specifies the complexity of the hypothesis class $\mathcal{H}_\infty$ (Caponnetto & De Vito, 2007); a faster eigen-decay (large $\xi$) implies the small complexity of the class. The assumption (A3) controls the smoothness of the target functions $\phi_0, \phi_1$. In fact, the functions are included in $\mathcal{H}_\infty$, since $\phi_i \in \mathcal{R}(\Sigma_\infty^{1/2}) \subset \mathcal{H}_\infty$. When a function $f$ has the expansion $f = \sum_\ell a_\ell \psi_\ell$, the assumption $f \in \mathcal{R}(\Sigma_\infty^r)$ means $a_\ell = o(\ell^{-\xi r - 1/2})$. A function with a larger $r$ is smoother, which is easier to learn. It is known (Caponnetto & De Vito, 2007; Nitanda & Suzuki, 2021) that $\xi$ and $r$ are the two basic parameters to control the convergence rate of generalization attained by kernel regression for a large sample size. Under the assumptions (A3) and (A4), given $N$ i.i.d. training data $(X_i, Y_i)$ with $X_i \sim \rho_X$ and $Y_i = \varphi_0(X_i) + \varepsilon_i$ with additive noise $\varepsilon_i \sim N(0, \sigma^2)$, the kernel ridge regression $\hat\varphi_\lambda$ with the regularization parameter $\lambda = N^{-\xi/(2r\xi+1)}$ achieves the generalization $E[\|\hat\varphi_\lambda - \varphi_0\|^2_{L^2(\rho_X)}] = O(N^{-2r\xi/(2r\xi+1)})$ for any function $\varphi_0$ with $\varphi_0 \in \mathcal{R}(\Sigma^r)$, and it is known this rate is optimal.

In the sequel, when $\phi \in \mathcal{R}(\Sigma^r)$ and $\phi = \Sigma^r \psi$, we write $\|\Sigma^{-r}\phi\| := \|\psi\|$.

### E.4 GENERALIZATION BOUND

The dynamical behavior of pre-training can be discussed exactly in the setting of Nitanda & Suzuki (2021). Let $\widehat\phi_0$ be the result of pre-training, i.e., $\widehat\phi_0 := g_{\overline\Theta_{pre}^{(T_0)}}$, where $\overline\Theta_{pre}^{(T_0)}$ is the averaged parameter by ASGD. By optimizing the regularization parameter $\lambda_0$, Corollary 1 in Nitanda & Suzuki (2021) shows that for sufficiently large $T_0$, with a choice of $\lambda_0 = T_0^{-\xi/(2r_0\xi+1)}$,

$$E\|\widehat\phi_0 - \phi_0\|^2_{L^2(\rho_X)} \leq \varepsilon_M + cT_0^{-\frac{2r_0\xi}{2r_0\xi+1}}\left(1 + \|\Sigma_\infty^{-r_0}\phi_0\|^2_{L^2(\rho_X)}\right) \tag{13}$$

with high probability, where $c$ is a universal constant and $\varepsilon_M$ can be arbitrarily small for a large $M$. As discussed in Section E.3, it is known (Caponnetto & De Vito, 2007) that the rate $T_0^{-\frac{2r_0\xi}{2r_0\xi+1}}$ achieves the minimax optimal rate with respect to $T_0$ over the class specified by $r_0$ and $\xi$.

In fine-tuning, the initial parameter is given by $\Theta^{(0)} = \overline\Theta_{pre}^{(T_0)}$, and ASGD is applied with $(x_t, y_t)$ for $t = 1, \ldots, T_1$.

By extending Theorem 1 in Nitanda & Suzuki (2021), we can derive a generalization bound in the following theorem. Recall that the regularization coefficient and learning rate of fine-tuning are denoted by $\lambda_1$ and $\eta_1$, respectively.

**Theorem 2.** *Suppose Assumptions (A1)-(A3) hold. After pre-training that gives Eq. (13), fine-tune the network by Eq. (9) with a learning rate $\eta_1$ and regularization coefficient $\lambda_1$ that satisfy $\|\Sigma_\infty\|_{op} \geq \lambda_1 > 0$ and $4(6 + \lambda_1)\eta_1 \leq 1$. Then, for any $\varepsilon > 0$, $\delta \in (0,1)$, and $T_1 \in \mathbb{N}$, there exists $M_0 \in \mathbb{N}$ such that for any $M \geq M_0$, the following bound holds with probability at least $1 - \delta$ over the random initialization of pre-training:*

$$E\|g_{\overline\Theta^{(T_1)}} - \varphi\|^2_{L^2(\rho_X)}$$
$$\leq \varepsilon + c_0\lambda_1^{2r_0}\|\Sigma_\infty^{-r_0}\phi_0\|^2_{L^2(\rho_X)} + c_1\lambda_1^{2r_1}\|\Sigma_\infty^{-r_1}\phi_1\|^2_{L^2(\rho_X)}$$
$$+ \frac{c_2}{T_1 + 1}\left\{\lambda_1^{-1}E\|\widehat\phi_0 - \phi_0\|^2_{L^2(\rho_X)}\left(1 + \|\Sigma_\infty^{-r_0}\phi_0\|^2_{L^2(\rho_X)}\right) + \lambda_1^{2r_1-1}\|\phi_0\|^2_{L^2(\rho_X)} + \|\phi_1\|^2_{\mathcal{H}_\infty}\right\}$$
$$+ \frac{c_3}{(T_1+1)^2\eta_1^2}\left\{\lambda_1^{-2}E\|\widehat\phi_0 - \phi_0\|^2_{L^2(\rho_X)}\left(1 + \|\Sigma_\infty^{-r_0}\phi_0\|^2_{L^2(\rho_X)}\right) + \lambda_1^{2r_1-2}\|\phi_0\|^2_{L^2(\rho_X)} + \lambda_1^{-1}\|\phi_1\|^2_{\mathcal{H}_\infty}\right\}$$
$$+ \frac{c_4}{T_1 + 1}\left(1 + \|\varphi\|^2_{\mathcal{H}_\infty} + 24\|\Sigma_\infty^{-r_0}\varphi\|^2_{L^2(\rho_X)}\right)\text{Tr}\left[\Sigma_\infty(\Sigma_\infty + \lambda_1 I)^{-1}\right], \tag{14}$$

where $\widehat{\phi}_0$ *is the result of pre-training and* $c_i$ $(i = 0, 1, 2, 3, 4)$ *are universal constants.*

The term $\varepsilon$ is arbitrarily small for a large value of $M$, i.e., wide network. The proof of Theorem 2 will be given in Section E.6.

### E.5 ANALYSIS OF CONVERGENCE RATES

We consider the rates of the generalization bound for $E\|g_{\overline{\Theta}^{(T_1)}} - \varphi\|^2_{L^2(\rho_X)}$ with respect to $T_1$. As typical cases, we assume $\lambda_1 \to 0$ and $\eta_1 = O(1)$ as $T_1 \to \infty$. The dominant terms in Eq. (14) may vary according to the configurations of $\lambda_1$ and $\eta_1$ with respect to $T_1$. We will show the rates in some settings that are relevant to transfer learning.

First, note that under Assumption (A4), the factor $\mathrm{tr}[\Sigma_\infty(\Sigma_\infty + \lambda_1 I)^{-1}]$ is given by (Caponnetto & De Vito, 2007)

$$\mathrm{Tr}\left[\Sigma_\infty(\Sigma_\infty + \lambda_1 I)^{-1}\right] = O(\lambda_1^{-1/\xi}). \tag{15}$$

By neglecting $\varepsilon$, the terms in Eq. (14) thus have the following rates:

$$(a0) \; \lambda_1^{2r_0}, \qquad (a1) \; \lambda_1^{2r_1}, \qquad (b) \; T_1^{-1}\lambda_1^{-1}R_0, \qquad (c) \; T_1^{-1}\lambda_1^{2r_1-1}, \qquad (d) \; T_1^{-1},$$

$$(e) \; T_1^{-2}\eta_1^{-2}\lambda_1^{-2}R_0, \quad (f) \; T_1^{-2}\eta_1^{-2}\lambda_1^{2r_1-2} \quad (g) \; T_1^{-2}\eta_1^{-2}\lambda_1^{-1}, \quad (h) \; T_1^{-1}\lambda_1^{-1/\xi}. \tag{16}$$

Here $R_0 := E\|\widehat{\phi}_0 - \phi_0\|^2_{L^2(\rho_X)}$ is of constant rate with respect to $T_1$, but explicitly shown for the later use.

Since $\lambda_1 \to 0$ and $r_1 \geq 1/2$, the terms (c) and (d) are of smaller rate than (b). Likewise, (f) and (g) are smaller than (e). The candidates of dominant terms are thus (a0), (a1), (b), (e), (g), and (h).

#### E.5.1 LARGE REGULARIZATION COEFFICIENT

In transfer learning, it is reasonable to use strong regularization in fine-tuning, which encourages the parameters to stay close to the initial value that is obtained in the pre-training. In this subsection, we consider the case where $\lambda_1$ is larger than $T_1^{-\frac{\xi}{2r_1\xi+1}}$, which would be the optimal rate if the network was trained with random initialization (see Nitanda & Suzuki (2021)). If $\lambda_1 = T_1^{-\frac{\xi}{2r_1\xi+1}}$ was taken, it is easy to see that the influence of pre-training would not appear explicitly in the convergence rate. In the sequel, we write $F \ll G(T_1)$ if there are $a > 0$ and $T^*$ such that $F \leq aG(T_1)$ for all $T_1 \geq T^*$. In this notation, we assume

$$\lambda_1 \gg T_1^{-\frac{\xi}{2r_1\xi+1}}. \tag{17}$$

The rate $\lambda_1 = T_1^{-\frac{\xi}{2r_1\xi+1}}$ is given by equating the rates of (a1) and (h). Therefore, under the assumption of Eq. (17), the rate (a1) is larger than (h), and thus it suffices to consider (a0), (a1), (b) and (e) as the candidates of dominant terms. Note that, if $\lambda_1 \to 0$, the terms (a0) and (a1) decrease, while (b) and (e) increase to infinity.

We will discuss below the possible cases of dominant terms under the assumption Eq. (17). In the analysis, although the error of the pre-training $R_0$ is regarded as a constant, we yet wish to consider the dependence of the fine-tuning result on $R_0$. We thus set the regularization coefficient $\lambda_1$ dependent on $R_0$, and show that in all the cases, the generalization bound takes the form

$$E\|g_{\overline{\Theta}^{(T_1)}} - \varphi\|^2_{L^2(\rho_X)} \leq \varepsilon + CR_0^\nu T_1^{-\beta}, \tag{18}$$

where $\nu > 0$ is a constant. As $R_0 \leq c' + A'T_0^{-\alpha'}$ from Eq. (13), the factor $R_0^\alpha$ can be bounded from above as

$$R_0^\nu \leq c + AT_0^{-\alpha'\nu}$$

for large $T_0$. As a result, we obtain

$$E\|g_{\overline{\Theta}^{(T_1)}} - \varphi\|^2_{L^2(\rho_X)} \leq \varepsilon + C(c + AT_0^{-\alpha})T_1^{-\beta}, \tag{19}$$

where $C, c, A$ are constants. As we will see, the exponents $\alpha$ and $\beta$ depend on $r_0, r_1, \xi$ and $\eta_1$. Eq. (19) accords with the bound in Theorem 1.

In the sequel, we use $\zeta \geq 0$ for the learning rate such that $\eta_1 = T_1^{-\zeta}$.

**Case I: Small learning rate $T_1 \lambda_1 \eta_1^2 \ll 1$.** In this case, (e) $\gg$ (b). We equate (a0) or (a1) with (e) to obtain $\lambda_1$ for achieving the best possible upper bound of the two terms.

**(I-A) $r_0 \geq r_1$.** Since $\lambda_1 \to 0$ for $T_1 \to \infty$, (a1) is larger than (a0). By equating (a1) and (e), we find that the best choice of $\lambda_1$ is

$$\lambda_1 = O\big(T_1^{-\frac{1-\zeta}{r_1+1}}\big).$$

We further consider

$$\lambda_1 = T_1^{-\frac{1-\zeta}{r_1+1}} R_0^\nu$$

for dependence on $R_0$. To determine $\nu$, we assume that $R_0$ is a small value, and consider the rate of (a1) and (e) with respect to $R_0$ after plugging the above $\lambda_1$ to them. By equating the rates of (a1) $T_1^{-\frac{2r_1(1-\zeta)}{r_1+1}} R_0^{2r_1\nu}$ and (e) $T_1^{-\frac{2r_1(1-\zeta)}{r_1+1}} R_0^{1-2\nu}$, the best possible rate of $R_0$ is attained by $\nu = 1/(2r_1 + 2)$. The dominant rate of Eq. (16) is thus

$$T_1^{-\frac{2r_1(1-\zeta)}{r_1+1}} R_0^{\frac{r_1}{r_1+1}} \tag{20}$$

attained by

$$\lambda_1 = T_1^{-\frac{1-\zeta}{r_1+1}} R_0^{\frac{1}{2(r_1+1)}} \tag{21}$$

We need to identify the conditions on $\zeta_1$ to meet the requirements. The condition $\lambda_1 \to 0$ is equivalent to $\zeta < 1$. There are two other conditions: $\lambda_1 \gg T_1^{-\frac{\xi}{2r_1\xi+1}}$ and $T_1 \lambda_1 \eta_1^2 \ll 1$. Given $R_0$ is of constant rate, the former is equivalent to $-\frac{\xi}{2r_1+1} \leq -\frac{1-\zeta}{1+r_1}$, which results in

$$\zeta \geq \frac{r_1\xi + 1 - \xi}{2r_1\xi + 1}.$$

The latter condition is equivalent to $1 - 2\zeta - \frac{1-\zeta}{r_1+1} \leq 0$, which is

$$\zeta \geq \frac{r_1}{2r_1 + 1}.$$

It is not difficult to see

$$\frac{r_1}{2r_1 + 1} > \frac{r_1\xi + 1 - \xi}{2r_1\xi + 1}$$

for $\xi > 1$. As a result, the condition on $\zeta$ is

$$\frac{r_1}{2r_1 + 1} \leq \zeta < 1. \tag{22}$$

If $\eta_1 = T_1^{-\zeta}$ is taken to satisfy this condition, the optimal rate of $\lambda_1$ is given by Eq. (21). Finally, the resulting generalization bound is given by

$$E\|g_{\overline{\Theta}^{(T_1)}} - \varphi\|_{L^2(\rho_X)}^2 \leq \varepsilon + c T_1^{-\frac{2r_1(1-\zeta)}{r_1+1}} R_0^{\frac{r_1}{r_1+1}}. \tag{23}$$

**(I-B) $r_1 > r_0$:** In this case, (a0) is of larger rate than (a1). By a similar argument to (I-A), with the rate

$$\lambda_1 = T_1^{-\frac{1-\zeta}{r_0+1}} R_0^{\frac{1}{2r_0+2}}, \tag{24}$$

The generalization bound is given by

$$E\|g_{\overline{\Theta}^{(T_1)}} - \varphi\|_{L^2(\rho_X)}^2 \leq \varepsilon + c T_1^{-\frac{2r_0(1-\zeta)}{r_0+1}} R_0^{\frac{r_0}{r_0+1}}. \tag{25}$$

The condition on $\zeta$ is

$$\max\left\{\frac{r_0}{2r_0 + 1}, \frac{(2r_1 - r_0)\xi + 1 - \xi}{2r_1\xi + 1}\right\} \leq \zeta < 1. \tag{26}$$

| $r_0, r_1$ | $\eta_1 = T^{-\zeta}$ | Bound |
|---|---|---|
| $r_0 \geq r_1$ | $\frac{r_1}{2r_1+1} \leq \zeta < 1$ | $\varepsilon + c'\left(c + T_0^{-\frac{2r_0\xi}{2r_0\xi+1}\cdot\frac{r_1}{r_1+1}}\right)T_1^{-\frac{2r_1(1-\zeta)}{r_1+1}}$ |
| $r_0 < r_1$ | $\max\left\{\frac{r_0}{2r_0+1}, \frac{(2r_1-r_0)\xi+1-\xi}{2r_1\xi+1}\right\} \leq \zeta < 1$ | $\varepsilon + c'\left(c + T_0^{-\frac{2r_0\xi}{2r_0\xi+1}\cdot\frac{r_0}{r_0+1}}\right)T_1^{-\frac{2r_0(1-\zeta)}{r_0+1}}$ |
| $r_0 \geq r_1$ | $\zeta \leq \frac{r_1}{2r_1+1}$ | $\varepsilon + c'\left(c + T_0^{-\frac{2r_0\xi}{2r_0\xi+1}\cdot\frac{2r_1}{2r_1+1}}\right)T_1^{-\frac{2r_1}{2r_1+1}}$ |
| $r_0 < r_1 \leq r_0 + \frac{\xi-1}{2\xi}$ | $\zeta \leq \frac{r_0}{2r_0+1}$ | $\varepsilon + c'\left(c + T_0^{-\frac{2r_0\xi}{2r_0\xi+1}\cdot\frac{2r_0}{2r_0+1}}\right)T_1^{-\frac{2r_0}{2r_0+1}}$ |

Table 1: Generalization bounds in various conditions.

**(Case II): large learning rate** $T_1\lambda_1\eta_1^2 \gg 1$. Next, we consider the case where the learning rate $\eta_1$ is large so that $T_1\lambda_1\eta_1^2 \gg 1$, which includes the constant $\eta_1$. Under this condition, (b) is of larger rate than (e).

**(II-A)** $r_0 \geq r_1$. In this case, (a1) is of larger rate than (a0). A similar argument to (I-A) provides

$$\lambda_1 = T_1^{-\frac{1}{2r_1+1}} R_0^{\frac{1}{2r_1+1}}, \tag{27}$$

and the generalization bound is given by

$$E\|g_{\overline{\Theta}^{(T_1)}} - \phi_0\|_{L^2(\rho_X)}^2 \leq \varepsilon + cT_1^{-\frac{2r_1}{2r_1+1}} R_0^{\frac{2r_1}{2r_1+1}}. \tag{28}$$

The conditions are $\lambda_1 \gg T_1^{-\frac{\xi}{2r_1\xi+1}}$ and $T_1\lambda_1\eta_1^2 \gg 1$. The former condition always holds for $\xi > 1$, and the latter is equivalent to $\zeta \leq \frac{r_1}{2r_1+1}$. The resulting condition on $\zeta$ is

$$0 < \zeta \leq \frac{r_1}{2r_1+1}. \tag{29}$$

**(II-B)** $r_1 > r_0$**:** In this case, (a0) is of larger rate. With

$$\lambda_1 = T_1^{-\frac{1}{2r_0+1}} R_0^{\frac{1}{2r_0+1}}, \tag{30}$$

the generalization bound is given by

$$E\|g_{\overline{\Theta}^{(T_1)}} - \phi_0\|_{L^2(\rho_X)}^2 \leq \varepsilon + cT_1^{-\frac{2r_0}{2r_0+1}} R_0^{\frac{2r_0}{2r_0+1}}. \tag{31}$$

The conditions $\lambda_1 \gg T_1^{-\frac{\xi}{2r_1\xi+1}}$ and $T_1\lambda_1\eta_1^2 \gg 1$ are respectively $r_1 \leq r_0 + \frac{\xi-1}{2\xi}$ and $\zeta \leq \frac{r_0}{2r_0+1}$. Thus, we require

$$0 < \zeta \leq \frac{r_0}{2r_0+1}, \qquad r_0 < r_1 \leq r_0 + \frac{\xi-1}{2\xi}, \tag{32}$$

In summary, the generalization bounds in various conditions are summarized in Table 1.

E.6 PROOF OF THEOREM 2

The proof of Theorem 2 is based on the application of the theory in Nitanda & Suzuki (2021) to the fine-tuning phase, adapting the initialization given by the result of pre-training $\widehat{\phi}_0$.

In the sequel, we focus on the fine-tuning with $T_1$ samples with $y_t = \varphi(x_t)$. Recall that

$$\varphi(x) = \phi_0(x) + \phi_1(x).$$

### E.6.1 REFERENCE ASGD ON RKHS

We use a surrogate sequence of functions in an RKHS for the proof. Let $k_M$ be the random feature approximation of the TNK $k_\infty$, i.e.,

$$k_M(x, x') = \frac{1}{M} \sum_{r=1}^{M} \sigma(b_r^T x)\sigma(b_r^T x') + \frac{x^T x'}{M} \sum_{r=1}^{M} \sigma'(b_r^T x)\sigma'(b_r^T x'), \tag{33}$$

where $(b_r)_{r=1}^{M}$ is i.i.d. random sample from the uniform distribution on the unit sphere $\mathbb{S}^{d-1}$. The associated RKHS is denoted by $\mathcal{H}_M$.

A reference ASGD is defined by the following update rule of functions in the RKHS $\mathcal{H}_M$:

$$g^{(t+1)} = (1 - \eta\lambda)g^{(t)} - \eta(g^{(t)}(x_t) - y_t)k_M(\cdot, x_t), \qquad (t = 0, \dots, T-1) \tag{34}$$

with the initialization given by $g^{(0)} := \widehat{\phi}_0$. The average is taken at the final step:

$$\bar{g}^{(T_1)} := \frac{1}{T_1 + 1} \sum_{t=0}^{T_1} g^{(t)}. \tag{35}$$

By considering continual learning of pre-training and fine-tune, a slight modification of (Nitanda & Suzuki, 2021, Propososion A) derives the following proposition.

**Proposition 3.** *Assume (A1) and (A2). Suppose that $\eta_1\lambda_1 < 1$. Then for any $T_1 \in \mathbb{N}$ and $\varepsilon > 0$, there is $M_* = M_*(T_1, \varepsilon) \in \mathbb{N}$ such that during the fine-tuning learning*

$$\|\bar{g}^{(t)} - g_{\overline{\Theta}^{(t)}}\|_{L^\infty(\rho_X)} \le \varepsilon, \tag{36}$$

*holds for any $M \ge M_*$ and $0 \le t \le T_1$.*

This proposition shows that, if we use a very wide network, the learning of ASGD in the parameter space can be approximated by the reference ASGD on the RKHS with negligible error.

The generalization bound will be given by the following decomposition:

$$\|g_{\overline{\Theta}^{(T_1)}} - \varphi\|_{L^2(\rho_X)}^2 \le 2\|g_{\overline{\Theta}^{(T_1)}} - \bar{g}^{(T_1)}\|_{L^2(\rho_X)}^2 + 2\|\bar{g}^{(T_1)} - \varphi\|_{L^2(\rho_X)}^2, \tag{37}$$

in which the first term of the right hand side is bounded by Proposition 3 with an arbitrary small value $\varepsilon$ for large $M$. The second term will be discussed in the next subsection.

### E.6.2 CONVERGENCE RATES OF REFERENCE ASGD

In this section, we write $\lambda$ and $\eta$ for $\lambda_1$ and $\eta_1$ for simplicity. The covariance operators $\Sigma_\infty$ and $\Sigma_M$ for $\mathcal{H}_\infty$ and $\mathcal{H}_M$, respectively, are defined by

$$\begin{aligned} \Sigma_\infty &:= E_{\rho_X}[k_\infty(\cdot, X) \otimes k_\infty(\cdot, X)^*], \\ \Sigma_M &:= E_{\rho_X}[k_M(\cdot, X) \otimes k_M(\cdot, X)^*], \end{aligned} \tag{38}$$

where $*$ denotes the adjoint; equivalently,

$$\Sigma_\infty f = \int k_\infty(\cdot, x)f(x)d\rho_X(x), \quad \Sigma_M h = \int k_M(\cdot, x)h(x)d\rho_X(x),$$

for $f \in \mathcal{H}_\infty, h \in \mathcal{H}_M$. The regularized target functions $\phi_{M,\lambda}^{(i)}$ $(i = 0, 1)$ are defined by

$$\phi_{M,\lambda}^{(i)} := (\Sigma_M + \lambda I)^{-1}\Sigma_M \phi_i \qquad (i = 0, 1). \tag{39}$$

$\phi_{\infty,\lambda}^{(i)}$ is defined similarly with $\Sigma_\infty$. Note that $\varphi_{M,\lambda} := (\Sigma_M + \lambda I)^{-1}\Sigma_M \varphi = \phi_{M,\lambda}^{(0)} + \phi_{M,\lambda}^{(1)}$.

First, we decompose $\|\bar{g}^{(T_1)} - \varphi\|_{L^2(\rho_X)}^2$ by

$$\begin{aligned} \|\bar{g}^{(T_1)} - \varphi\|_{L^2(\rho_X)}^2 &= \|\bar{g}^{(T_1)} - \varphi_{M,\lambda} + \phi_{M,\lambda}^{(0)} + \phi_{M,\lambda}^{(1)} - \phi_0 - \phi_1\|_{L^2(\rho_X)}^2 \\ &\le 3\|\bar{g}^{(T_1)} - \varphi_{M,\lambda}\|_{L^2(\rho_X)}^2 + 3\|\phi_{M,\lambda}^{(0)} - \phi_0\|_{L^2(\rho_X)}^2 + 3\|\phi_{M,\lambda}^{(1)} - \phi_1\|_{L^2(\rho_X)}^2. \end{aligned}$$

The second and third terms are known to have a bound, with high probability, (Nitanda & Suzuki, 2021, Propositions C and D)

$$\|\phi_{M,\lambda}^{(i)} - \phi_i\|_{L^2(\rho_X)}^2 \leq \varepsilon + \lambda^{2r_i}\|\Sigma_\infty^{-r_i}\phi_i\|_{L^2(\rho_X)}^2, \qquad (i = 0, 1), \tag{40}$$

where $\varepsilon$ is arbitrarily small for large $M$. We have thus, with high probability,

$$\|\bar{g}^{(T_1)} - \varphi\|_{L^2(\rho_X)}^2 \leq \varepsilon + 3\|\bar{g}^{(T_1)} - \varphi_{M,\lambda}\|_{L^2(\rho_X)}^2 + 3\lambda^{2r_0}\|\Sigma_\infty^{-r_0}\phi_0\|_{L^2(\rho_X)}^2 + 3\lambda^{2r_1}\|\Sigma_\infty^{-r_1}\phi_1\|_{L^2(\rho_X)}^2. \tag{41}$$

As shown in Nitanda & Suzuki (2021), the term $\|\bar{g}^{(T_1)} - \varphi_{M,\lambda}\|_{L^2(\rho_X)}^2$ can be analyzed by the bias and noise terms of the stochastic recursion on RKHS Eq. (34), which is rewritten as

$$g^{(t+1)} = (I - \eta H_t - \eta\lambda I)g^{(t)} + \eta y_t k_M(\cdot, x_t), \tag{42}$$

where

$$H_t := k_M(\cdot, x_t) \otimes k_M(\cdot, x_t)^*,$$

is a one-sample estimate of $\Sigma_M$. By subtracting $\varphi_{M,\lambda}$ from both hand sides of Eq. (42), we have

$$g^{(t+1)} - \varphi_{M,\lambda} = (I - \eta H_t - \eta\lambda I)(g^{(t)} - \varphi_{M,\lambda}) + \beta_t, \tag{43}$$

where

$$\beta_t = \eta y_t k_M(\cdot, x_t) - \eta(H_t + \lambda I)(\Sigma_M + \lambda I)^{-1}\Sigma_M\varphi_{M,\lambda}$$

is the zero mean noise term. Using this recursive formula, Nitanda & Suzuki (2021) derives a bound:

$$\begin{aligned}
\|\bar{g}^{(T)} - \varphi_{M,\lambda}\|_{L^2(\rho_X)}^2 &\leq \frac{c_1}{T_1 + 1}\|(\Sigma_M + \lambda I)^{-1/2}(g^{(0)} - \varphi_{M,\lambda})\|_{L^2(\rho_X)}^2 \\
&+ \frac{c_2}{(T_1 + 1)^2\eta^2}\|(\Sigma_M + \lambda I)^{-1}(g^{(0)} - \varphi_{M,\lambda})\|_{L^2(\rho_X)}^2 \\
&+ \frac{c_3}{T_1 + 1}\left(1 + \|\varphi\|_{L^2(\rho_X)}^2 + 24\|\Sigma_\infty^{-r_1}\varphi\|_{L^2(\rho_X)}^2\right)\mathrm{Tr}[\Sigma_M(\Sigma_M + \lambda I)^{-1}].
\end{aligned} \tag{44}$$

To bound this expression further, we use Proposition B in Nitanda & Suzuki (2021)

$$\|(\Sigma_M + \lambda I)^{-1/2}\phi_{M,\lambda}^{(i)}\|_{L^2(\rho_X)}^2 \leq 2\|\phi_i\|_{\mathcal{H}_\infty}^2. \tag{45}$$

Then, using the decomposition $g^{(0)} - \varphi_{M,\lambda} = (\hat{\phi}_0 - \phi_0) + (\phi_0 - \phi_{M,\lambda}^{(0)}) - \phi_{M,\lambda}^{(1)}$, we obtain that, with high probability,

$$\begin{aligned}
&\|(\Sigma_M + \lambda I)^{-1/2}(g^{(0)} - \varphi_{M,\lambda})\|_{L^2(\rho_X)}^2 \\
&\leq 3\|(\Sigma_M + \lambda I)^{-1/2}(\hat{\phi}_0 - \phi_0)\|_{L^2(\rho_X)}^2 + 3\|(\Sigma_M + \lambda I)^{-1/2}(\phi_0 - \phi_{M,\lambda}^{(0)})\|_{L^2(\rho_X)}^2 \\
&\quad + 3\|(\Sigma_M + \lambda I)^{-1/2}\phi_{M,\lambda}^{(1)}\|_{L^2(\rho_X)}^2 \\
&\leq \frac{3}{\lambda}\|\hat{\phi}_0 - \phi_0\|_{L^2(\rho_X)}^2 + \frac{3}{\lambda}\lambda^{2r_0}\|\phi_0\|_{L^2(\rho_X)}^2 + \varepsilon_M + 6\|\phi_1\|_{\mathcal{H}_\infty}^2,
\end{aligned} \tag{46}$$

where we use Eq. (40) for the second and third terms and Eq. (45) for the fourth term in the last inequality. Similarly, with high probability, we have

$$\begin{aligned}
&\|(\Sigma_M + \lambda I)^{-1}(g^{(0)} - \varphi_{M,\lambda})\|_{L^2(\rho_X)}^2 \\
&\leq 3\|(\Sigma_M + \lambda I)^{-1}(\hat{\phi}_0 - \phi_0)\|_{L^2(\rho_X)}^2 + 4\|(\Sigma_M + \lambda I)^{-1}(\phi_0 - \phi_{M,\lambda}^{(0)})\|_{L^2(\rho_X)}^2 \\
&\quad + 3\|(\Sigma_M + \lambda I)^{-1}\phi_{M,\lambda}^{(1)}\|_{L^2(\rho_X)}^2 \\
&\leq \frac{3}{\lambda^2}\|\hat{\phi}_0 - \phi_0\|_{L^2(\rho_X)}^2 + \frac{3}{\lambda^2}\lambda^{2r_0}\|\phi_0\|_{L^2(\rho_X)}^2 + \varepsilon_M + \frac{6}{\lambda}\|\phi_1\|_{\mathcal{H}_\infty}^2.
\end{aligned} \tag{47}$$

It is also known (Nitanda & Suzuki, 2021, Proposition B) that, for $\lambda \leq \|\Sigma_\infty\|$,

$$\mathrm{Tr}[\Sigma_M(\Sigma_M + \lambda I)^{-1}] \leq 3\mathrm{Tr}[\Sigma_\infty(\Sigma_\infty + \lambda I)^{-1}]. \tag{48}$$

Combining Eqs.(41), (44), (46), (47), and (48), we obtain the assertion of the theorem.

