# OpenReview forum: "A Scaling Law for Syn-to-Real Transfer: How Much Is Your Pre-training Effective?"
_ICLR.cc/2022/Conference — ICLR 2022 Submitted_

### Official Review · Reviewer_EiPP · 2021-10-27

**Correctness:** 3
**Technical Novelty And Significance:** 2
**Empirical Novelty And Significance:** 2
**Recommendation:** 5
**Confidence:** 3

**Main Review:**

Strenght

- The paper proposes a scaling law for transfer learning that takes into account the size of the pretraining dataset.
- The experimental analysis is very good and cover very important tasks for the CV community  (e.g.  object detecction, segmentation, normal estimation)
- The authors provide some justification of the derived law.

> Updated after rebuttal.

Weakness.
- Major Concern #1. I consider the sim2real aspect of this paper to be a bit overstated. In my understanding, the dervied scaling law can be used in any transfer learning  setting that involves pretraining and finetuning. Why is it limited to sim2real or what does it make it specific to the sim2real scenario? I understand the experiments are conducted with a synthetic source dataset but this shouldnt be the only reason.

>After rebuttal. This concern  remains. I agree of the tremendous importance of the sim2real transfer, but the derivations presented here are not specific to the sim2real scenario. The narrowing of the scope seems still a bit artificial and not necessary.

- Major Concern #2. Several times in the paper the authors refers to the coeficient C as the transfer gap (introduced in sec 3.3 without explanation). It is further mentioned than this is related to the rendering settings (also in sec 3.3). I didnt find anywhere any justification or intuition for this relationship. Just stating that C is the transfer gap seems a bit artifical, same for saying that this is related to the rendering settings. I could imagine two real datasets that are dissimilar (e.g. different camera parameters or black-white vs color) and this C may also be big (no rendering involved).

>After rebuttal. The concern partially remains. I agree that C is induced by the dissimilarity between pre-training and fine-tuning tasks, but attributing that to the rendering setting without further justification is not proper. There are many component that may be having a role in what  “ dissimilarity between pre-training and fine-tuning tasks” is. Without further justification, the claim seems very abstract.

- I am not completely sure what can we get from section 3.2 and Theorem 1 since the law was empirically determined, and again, in my understanding, nothing in this section is specific to the sim2real scenario.  Is there anything we can use from this theorem to justify C is the transfer gap and it is related to the rendering settings? This section is also introduced in a very fast mode with most of the material defer to the suppl. It also introduces a lot of notation in a very fast pace that could be confusing. I recommend the authors to explain better the connection between this and eq (2). Particularly with respect to the exponents \alpha and \beta. Is there anything else that we can get out of this theorem which was not  obtained experimentally? I also recomend the authors to add a regression task to their experimental setting since this is the one used to motivate this section.

> After rebuttal. This concern remains, I agree it is not trivial, but  in my opinion this is required to claim the connection between this section and the sim2real scenario. The paper title reads "A Scaling Law for Syn-to-Real Transfer" and this section goal seems to theoretically suport this claim, however,  in my understanding, nothing in this section is specific to the sim2real scenario. I am still not getting much  from section 3.2 and Theorem 1.

- More intuition on the exponents on the derived law (eq 2) should be provided \alpha, \betta \delta. We know that they are coeficient for sure, and that they are bounded below, but what are the pretraining and finetuning rates? What's the intuitive and/or theoretical interpretation of them?
> After rebuttal. Solved.

- I think the paper will also benefit from more discussion in section 3.1. For example, could the authors discussed other choices of g(n), showed the limitation and motivate why the chosen one is the most resonable one. Is there a way to "strongly connect" the choice of g(n) to the theoretical results from 3.2? Also will a better parametrization here help with the numerical inestability issues.
> After rebuttal. Solved.

- Major Concern #3 . I  feel what we can get out of this paper may be a bit limited. The idea of the transfer gap(or disimilarity between the source and target domains) being very important for transfer learning is not new. This has been extensively studied and also shown rigorously and experimental in more general settings. Similarly, it is well known than usually bigger models generalize better.

>After rebuttal. Partially remains. I agree on the importance but my concern is that the main take away here might be a bit limited. We know from before (without having the scaling law) that key for successful  transfer learning is the dissimilarity between source vs target. This is not new.

**Summary Of The Paper:**

This paper proposes a scaling law for transfer learning that takes into account  the size of both the pretraining and finetuning datasets. They showed the parameters of the scaling law can be  estimated and this can be used to approximate the performance of the model given the amount of pretraining data. The authors experiments are focused on the scenario where the pretranining dataset is synthetically generated.

**Summary Of The Review:**

Overall, I consider the paper contains some nice ideas and the experimental analysis to be interesting although  I  feel what we can get out of this paper may be a bit limited. I also believe that presentation of the work should be improved and the work would benefit from a major revision. I also recomend the authors to focus on the more general transfer learning scenario since the narrowing of the scope to a "sim-to-real scaling law" may seem a bit artificial.

---

> ### Author Response · Authors · 2021-11-18
> **Response**
>
> Thanks again for your insightful comments and feedback. Here, we will answer your questions in detail.
>
> > Q1. Major Concern #1. I consider the sim2real aspect of this paper to be a bit overstated. In my understanding, the dervied scaling law can be used in any transfer learning setting that involves pretraining and finetuning. Why is it limited to sim2real or what does it make it specific to the sim2real scenario? I understand the experiments are conducted with a synthetic source dataset but this shouldnt be the only reason.
>
> A1. As explained in our [earlier reply](https://openreview.net/forum?id=QhHMf5J5Jom&noteId=MPyh765UlgA), we believe that syn2real is the best scenario to derive the scaling law. It’s true that our results can be applied to general transfer learning, but it doesn’t mean our claim is overstated since syn2real is a special case of transfer learning.
>
> > Q2. Major Concern #2. Several times in the paper the authors refers to the coeficient C as the transfer gap (introduced in sec 3.3 without explanation). It is further mentioned than this is related to the rendering settings (also in sec 3.3). I didnt find anywhere any justification or intuition for this relationship. Just stating that C is the transfer gap seems a bit artifical, same for saying that this is related to the rendering settings. I could imagine two real datasets that are dissimilar (e.g. different camera parameters or black-white vs color) and this C may also be big (no rendering involved).
>
> A2. It seems there is some misunderstanding here. In this study, we intend to say that C is induced by the dissimilarity between pre-training and fine-tuning tasks. If we use the same data at the same task, C disappears and the standard power-law scaling appears. In syn2real, the rendering settings are one of the factors that controls the task similarity. The differences of the camera parameters and color settings should also be explained by the factors.
>
> > Q3. I am not completely sure what can we get from section 3.2 and Theorem 1 since the law was empirically determined, and again, in my understanding, nothing in this section is specific to the sim2real scenario. Is there anything we can use from this theorem to justify C is the transfer gap and it is related to the rendering settings?
>
> A3. Theorem 1 provides a double proof of the scaling law. Unfortunately, it is not easy to interpret C in the current derivation and it would be an interesting future direction．
>
> > Q4. More intuition on the exponents on the derived law (eq 2) should be provided \alpha, \betta \delta. We know that they are coeficient for sure, and that they are bounded below, but what are the pretraining and finetuning rates? What's the intuitive and/or theoretical interpretation of them?
>
> A4. We should have included more explanations in presenting eq.(2). The exponent $\beta$ determines the convergence rate with respect to fine-tuning data size.  From this viewpoint, $\delta(\gamma + n^{-\alpha})$ is the coefficient factor to the power law.  The influence of the pre-training appears in this coefficient, where the constant term $\delta\gamma$ comes from the irreducible loss of the pre-training task and $n^{-\alpha}$ expresses the effect of pre-training data size. The theoretical consideration in Sec. E.5 suggests that the rates $\alpha$ and $\beta$ can depend on both the target functions of pre-training and fine-tuning as well as the learning rate.
>
> > Q5. I think the paper will also benefit from more discussion in section 3.1. For example, could the authors discussed other choices of g(n), showed the limitation and motivate why the chosen one is the most resonable one.
>
> A5. A simple alternative would be g(n) = n^-\alpha. This version lacks \gamma, which means the pre-training effect is not satu	rated any more. We empirically compared this version and Eq.(1) in a new experiment (Section 4.4 and Figure 5 in the revised manuscript). The results clearly show the “no-saturation” law fails to capture the entire trend.

---

> ### Author Response · Authors · 2021-11-18
> **Response 2**
>
> > Q6. Major Concern #3 . I feel what we can get out of this paper may be a bit limited. The idea of the transfer gap(or disimilarity between the source and target domains) being very important for transfer learning is not new. This has been extensively studied and also shown rigorously and experimental in more general settings. Similarly, it is well known than usually bigger models generalize better.
>
> A6. Our main contribution is the discovery of the scaling law that takes into account the pre-training size. To the best of our knowledge, this kind of equation has never been studied before. Along with this, we have several findings here.
> - The scaling law tells us that C is the key whether syn2real transfer succeeds or not --- if C is small, increasing the pre-training size will pay off (Figure 3 (a)).
> - The scaling law predicts the final performance at a target task in a qualitative way (this is examined in our new experiments in Section 4.4 and Figure 5).
> - The scaling law holds for many different pairs of pre-training and fine-tuning. In NLP, the scaling behavior of English to Python transfer was studied by Hernandez et al. (2021), but similar behaviors have never been observed in computer vision.

---

> ### Comment · Reviewer_EiPP · 2021-11-23
> **Thank you for the response.**
>
> I thank the authors for the response and the updated revision.
> After reading the response and also other reviewers' feedback, several of my concerns still remain and therefore I have decided to keep my score. As before, I believe that the presentation of the work should be improved and the work would benefit from a major revision. I also recommend the authors focus on the more general transfer learning scenario since I still consider there is nothing in the current presentation that makes it specific to the sim-to-real problem.

---

> > ### Author Response · Authors · 2021-11-25
> > **Re**
> >
> > Thank you for your further comments.
> >
> > > After reading the response and also other reviewers' feedback, several of my concerns still remain and therefore I have decided to keep my score.
> >
> > We respect your decision, but could you be more specific about the concerns that still remain? We would like to know the problems (if exists) and fix them as much as possible.
> >
> > > I still consider there is nothing in the current presentation that makes it specific to the sim-to-real problem.
> >
> > In [our reply](https://openreview.net/forum?id=QhHMf5J5Jom&noteId=MPyh765UlgA), we did our best to explain why we focused on syn2real transfer. Indeed, Reviewer Ltke [admitted our point of view](https://openreview.net/forum?id=QhHMf5J5Jom&noteId=rQ6gkwr_hGG). If it still doesn't make sense to you, can you tell us which part of it you didn't agree with?

---

> > > ### Comment · Reviewer_EiPP · 2021-12-01
> > > **Re**
> > >
> > > I've detailed in my main review the list of concerns that still remain after rebuttal.
> > >
> > > I hope this helps to improve your paper.

---

### Official Review · Reviewer_9onL · 2021-11-02

**Correctness:** 2
**Technical Novelty And Significance:** 2
**Empirical Novelty And Significance:** 1
**Recommendation:** 3
**Confidence:** 3

**Main Review:**

Strengths:
- The paper explores an interesting relationship between pre-training dataset size and final accuracy on a downstream task after finetuning.

Weaknesses:
- The scaling law is derived by empirical observation.
- The fine-tuning dataset size is kept fixed in all experiments. This is, in my opinion, an oversimplification of the scaling law.
- The paper provides is no measurable quantitative evaluation of its results. The plots just show that the predicted points are "close" to the fitted line. How will the method compare with a different scaling law if these results are not provided?
- Applicability might be limited due to the possibility of just increasing the synthetic training set size until desired performances are reached.

Other:
- Figure 2 is not a log-log plot, as stated in sec. 3.1.

**Summary Of The Paper:**

The paper describes a scaling law for approximating the number of samples needed for pre-training to reach a certain performance on a downstream task. The experimental protocol evaluates the method by fitting the scaling law's parameters to a set of observation with least squares, though no quantitative evaluation is performed.

**Summary Of The Review:**

The paper tries to answer an interesting question (though practical interest might be limited) in estimating the number of samples needed for pre-training, however there are many simplifications (as also stated by the authors) and the empirical evaluations, on which the claim is based, lack measurable results.

---

> ### Author Response · Authors · 2021-11-18
> **Response**
>
> Thanks again for your insightful comments and feedback. Here, we will answer your questions in detail.
>
> > Q1. The scaling law is derived by empirical observation.
>
> A1. We derive the scaling law also in a theoretical way. Please see Theorem 1.
>
> > Q2. The fine-tuning dataset size is kept fixed in all experiments. This is, in my opinion, an oversimplification of the scaling law.
>
> A2. The fine-tuning data size was changed in the experiments in Figure 2. The full results are shown in Figure 9.
>
> > Q3. The paper provides is no measurable quantitative evaluation of its results. The plots just show that the predicted points are "close" to the fitted line. How will the method compare with a different scaling law if these results are not provided?
>
> A3. Thank you for the suggestion. We can measure the generalization performance by extrapolation, i.e., fitting the scaling law with small pre-training size n and making predictions on large n. We additionally conducted this experiment by comparing power-law scaling as a baseline and included the results in Section 4.4. In short, the results indicate that our scaling law (1) has the extrapolation ability (see Figure 5).
>
> > Q4. Applicability might be limited due to the possibility of just increasing the synthetic training set size until desired performances are reached.
>
> A4. This strategy (just increasing the number of synthetic images) fails when C is larger than the desired performance as shown in Figure 3 (c). It was discussed in Section 3.3.
>
> > Q5. Figure 2 is not a log-log plot, as stated in sec. 3.1.
>
> A5. Figure 2 is a log-log plot. Maybe it is not easy to recognize but the Y-axis is not evenly spaced.

---

> > ### Comment · Reviewer_9onL · 2021-11-23
> > **Response**
> >
> > Thanks for your response.
> > After reading this and other reviewers' responses I've decided to keep my original feedback.
> >
> > AA1.I still don't find a theoretical derivation of the scaling law: it doesn't seem that Theorem 1 is backed up by a proof. If it is, I am missing it.
> > AA2. There are some figures that graphically hint at what happens with different finetuning dataset sizes, but they are hard to measure. The finetuning dataset size is still not included in the scaling law.
> > AA4. But can this situation be predicted? And if the answer is positive, how many synthetic samples are needed to predict C with reasonable accuracy? The paper is not answering these questions because, related to Q3, there is no quantitative benchmark.
> >
> > Overall I feel that the paper might not be ready yet. I am not convinced by how the scaling law is derived and I suggest to implement a benchmark task in order to measure the effectiveness and accuracy of the proposed law for both research and application.

---

> > > ### Author Response · Authors · 2021-11-25
> > > **Re**
> > >
> > > Thank you for your additional suggestions.
> > >
> > > > AA1.I still don't find a theoretical derivation of the scaling law: it doesn't seem that Theorem 1 is backed up by a proof. If it is, I am missing it.
> > >
> > > The proof of Theorem 1 is given in Section E.5 (see Eq.(19)).  The location of the proof may not be clearly given in the main paper.  We will surely include it in the final version.
> > >
> > > > AA2. There are some figures that graphically hint at what happens with different finetuning dataset sizes, but they are hard to measure. The finetuning dataset size is still not included in the scaling law.
> > >
> > > We are afraid we don't understand what you exactly mean here. The right panel of Figure 9 shows how the test error behaves w.r.t. the fine-tuning data size, which clearly shows the power-law scaling (i.e. the error is decreasing linearly in log-log scale). We believe this is the opposite situation of "The finetuning dataset size is still not included in the scaling law".
> > >
> > > > AA4. But can this situation be predicted? And if the answer is positive, how many synthetic samples are needed to predict C with reasonable accuracy? The paper is not answering these questions because, related to Q3, there is no quantitative benchmark.
> > >
> > > Note that we cannot measure the estimation accuracy of C, because we cannot know the true C even in the toy examples. Instead, as replied in A3, we evaluated the extrapolation ability, which indicates at least 5 points (pre-training data size n = 2500, 5000, 10000, 20000, 40000) are sufficient to estimate the scaling law where the RMSE of the prediction is less than 0.003.
> > >
> > > > Overall I feel that the paper might not be ready yet. I am not convinced by how the scaling law is derived and I suggest to implement a benchmark task in order to measure the effectiveness and accuracy of the proposed law for both research and application.
> > >
> > > We respect your suggestion, but we cannot imagine what kind of a benchmark task you are expecting. Can you provide a specific example? Also, what does "the effectiveness and accuracy of the proposed law" exactly mean? What are the definitions of effectiveness and accuracy here?

---

### Official Review · Reviewer_Ltke · 2021-11-03

**Correctness:** 3
**Technical Novelty And Significance:** 3
**Empirical Novelty And Significance:** 3
**Recommendation:** 6
**Confidence:** 3

**Main Review:**

Idea/Method
-----------------
Studying the existence of a scaling law for syn-to-real as a function of irreducible domain discrepancy error and data size is important and has great practical benefits. A well fit scaling law would help make informed decisions about whether pre-training data needs to be scaled or diversified for downstream performance improvements. The assumptions leading to the inductive scaling law formulation are sound and it empirically fits real transfer errors very well.

I wasn't able to tell why the authors decided to focus only on syn-to-real transfer since the theoretical derivation does not distinguish between real-to-real or syn-to-real transfer. This is a key question that goes completely undiscussed. Were there negative results with real-to-real transfer? Was syn-to-real chosen so pre-training data complexity could be controlled?

Experiments
-----------------
Overall, I feel the experiments are minimal to justify the paper's claims
- In Fig. 1 there is no extrapolation. The curve is fit to all the points shown. A key feature of a useful scaling law is the ability to extrapolate findings. Fig. 5 and Section 4.4. show results with larger pre-training data sizes but it is unclear whether the curves are extrapolated from Fig. 1 or were the result of fitting parameters again to the new data. In both cases, it'd also be good to compare the estimated values on extrapolated points to show whether the fit curve generalizes (and also show how many points are needed to get a good generalizable fit)
- The authors use different error metrics for different tasks to fit the scaling law. Why does this work? This is especially counterintuitive for object detection, where 1 - mean average precision is used as the error metric. Is there anything beyond empirical justification for why this works?
- The error rates for all tasks are in the high 80s and 90s. These are significantly far away from the current best syn-to-real generalization results. The irreducible error term C is estimated to be in the high 80s, which is nowhere close to realistic settings that a practitioner would be in. It would be interesting if the authors did one experiment with a more realistic syn-to-real setting, for example doing GTA-V -> ADE20K/Cityscapes semantic segmentation
- In all cases, the authors use only 1% of real data, so that the effect of pre-training can be shown. This is again an artificial constraint that could be relieved by using more realistic synthetic data (such as GTA-V data), where the effect of pre-training is visible while using much higher % of real data

Related Work
-------------------
Related work is generally well written. In one part, the authors say, "Theoretical analysis was also attempted [Hutter, 2021][Bahri et al., 2021]". It'd be nice to discuss the differences there instead of just mentioning the papers.

Domain Adversarial Learning (https://arxiv.org/abs/1505.07818), a recent follow-up called f-Domain Adversarial Learning (https://arxiv.org/abs/2106.11344) and papers of similar flavour are quite related. They provide bounds on target domain risk as a function of source domain risk and domain discrepancy. They also provide practical algorithms for syn-to-real training. These works are missing in the related work discussion and I believe they warrant a discussion.

Writing
---------
Overall, the paper is well written. Minor typos here and there, such as in the 3rd line of the last para of related work. Figure 2. caption is opposite of the figure.

**Summary Of The Paper:**

This paper postulates a scaling law for pre-training and transfer learning via fine-tuning. They empirically postulate a law and also theoretically show that in a simpler setting with assumptions, a similar law should hold. The paper is written and experiments are done in a synthetic to real transfer setting. Experiments are done by pre-training on a synthetic dataset made using BlenderProc and transferring to real data ranging classification, object detection and semantic segmentation. They find that parameters of their scaling law can be estimated to find very good fits to empirical error across all the different transfer tasks.

**Summary Of The Review:**

Overall, I am positive about this paper. The method is well motivated and justified. I believe the experiments could be much stronger, especially since the experiments with real datasets would still be considered toy experiments due to the choice of pre-training data and % of real data used. It is hard to say whether these results would generalize, but I believe they warrant a discussion by the community.

---

> ### Author Response · Authors · 2021-11-18
> **Response**
>
> Thanks again for your insightful comments and feedback. Here, we will answer your questions in detail.
>
> > Q1. I wasn't able to tell why the authors decided to focus only on syn-to-real transfer since the theoretical derivation does not distinguish between real-to-real or syn-to-real transfer. This is a key question that goes completely undiscussed. Were there negative results with real-to-real transfer? Was syn-to-real chosen so pre-training data complexity could be controlled?
>
> A1. The main reason for focusing on syn2real setting is that the size of real data is critically limited. Please read our [earlier reply](https://openreview.net/forum?id=QhHMf5J5Jom&noteId=MPyh765UlgA). We didn’t conduct experiments in real-to-real settings, and we didn’t observe any negative results.
>
> > Q2. In Fig. 1 there is no extrapolation. The curve is fit to all the points shown. A key feature of a useful scaling law is the ability to extrapolate findings. Fig. 5 and Section 4.4. show results with larger pre-training data sizes but it is unclear whether the curves are extrapolated from Fig. 1 or were the result of fitting parameters again to the new data. In both cases, it'd also be good to compare the estimated values on extrapolated points to show whether the fit curve generalizes (and also show how many points are needed to get a good generalizable fit)
>
> A2. Thank you for the suggestion. We conducted this experiment and added the results in Section 4.4 and Figure 5. In short, the results indicate that our scaling law (1) has the extrapolation ability at least in objdet->objdet setting.
>
> > Q3. The authors use different error metrics for different tasks to fit the scaling law. Why does this work? This is especially counterintuitive for object detection, where 1 - mean average precision is used as the error metric. Is there anything beyond empirical justification for why this works?
>
> A3. Our claim is that the form of scaling law eq.(2) holds irrespective of the type of tasks as well as the loss or surrogate loss function, while the values of the parameters may depend on tasks and losses.  In the case of standard (non-transfer) learning, such a universal power law for the excess risk can be seen in many situations; see, e.g. [1] for regression and [2] for classification.  Also, the error in 0-1 loss by learning with surrogate loss is discussed in [3].
> We expect from the above existing theoretical results that the form of eq.(2) holds for a wide class, whereas the theoretical analysis in Sec. 3.2 and Sec. E discuss only the least square loss.
>
> References
> - [1] T. Suzuki (2020) Generalization bound of globally optimal non-convex neural network training: Transportation map estimation by infinite dimensional Langevin dynamics.  NeurIPS 2020.
> - [2] J.-Y. Audibert, A.B. Tsybakov (2007) Fast learning rates for plug-in classifiers. Ann. Stat. Vol. 35, No. 2, 608–633
> - [3] J. Zhang, T. Liu and D. Tao (2021) On the Rates of Convergence From Surrogate Risk Minimizers to the Bayes Optimal Classifier. IEEE Transactions on Neural Networks and Learning Systems, doi: 10.1109/TNNLS.2021.3071370.
>
>
> > Q4. In all cases, the authors use only 1% of real data, so that the effect of pre-training can be shown. This is again an artificial constraint that could be relieved by using more realistic synthetic data (such as GTA-V data), where the effect of pre-training is visible while using much higher % of real data
>
> A4. Figure 2 shows the cases where the fine-tuning data size was changed. The full results are shown in Figure 9.
>
> In addition, we argue that fine-tuning with a small amount of real data (e.g. 1%) is not artificial, but a practical setting. It is often difficult to prepare a sufficient amount of real data in a real task due to the limitations of images and annotations. Therefore, we believe that fine-tuning results on a small number of data is of practical value. We also discussed this topic at Section 3.3 (small fine-tuning v.s. big fine-tuning).
>
> > Q5. Related work
>
> A5. We elaborate on [Hutter, 2021][Bahri et al., 2021] as follows.
>
> \citet{hutter2021learning} analyzed a simple class of models that exhibits a power-law $n^{-\beta}$ in terms of data size $n$ with arbitrary $\beta > 0$. \citet{bahri2021explaining} addressed power laws under four regimes for model and data size. Note that these theoretical studies, unlike ours, are concerned with scaling laws in a non-transfer setting.
>
> We also include the domain adversarial studies in the new version (see the paragraph of transfer learning theory in Section 2).
>
> > Q6. Writing
>
> A6. We fixed the typos. Thanks!

---

> > ### Comment · Reviewer_Ltke · 2021-11-23
> > **Appreciate the updated work**
> >
> > Thanks for the detailed reply and additional experiments to show extrapolation. It is good to know that the law indeed extrapolates for 1 order or magnitude greater pre-training data size (at least in objdet->objdet setting). This was my major concern with the paper.
> >
> > I understand the authors' point of view for working on a scaling law for syn-to-real transfer. I believe this is a question that would naturally arise in a reader's mind and hence it should be discussed right off the bat. Right now, the amount of labelled data available for downstram tasks is used to motivate synthetic pre-training in the introduction, whereas the point brought up in the rebuttal talks about 1) the amount of real data available with permissive licenses and 2) the lack of a dataset with diverse tasks to enable researching a scaling law being the major reason for looking at syn-to-real generalization. I hope the discussion in the intro will be modified to include both of these motivations. Re: 2) the taskonomy (http://taskonomy.stanford.edu/) dataset could be something the authors could consider.
> >
> > Overall, I do agree with the other reviewers that trying to study whether the proposed law works in real->real setting would add a lot of value to this paper. The tasknomy dataset could be used as mentioned above. That said, I still believe that the paper in its current form has merits since synthetic->real training is being increasingly used in practice and a reliable scaling law would be useful for practitioners. Hence, I would still vote to accept the paper to allow it to be disccused at the conference.

---

> > > ### Author Response · Authors · 2021-11-25
> > > **Re**
> > >
> > > Thank you for your positive comments. Your feedback is really encouraging.
> > >
> > > > I hope the discussion in the intro will be modified to include both of these motivations.
> > >
> > > We will surely revise this part in the final version.
> > >
> > > > the taskonomy dataset
> > >
> > > Thank you for the suggestion. We considered using this dataset for syn2real once during the project, but decided not to use it because (1) all the scenes were limited to indoor scenes, and it would be difficult to transfer to real images including outdoor scenes such as COCO, and (2) the complexity of the images was uncontrollable. However, it would be beneficial for real2real. We will consider this idea for future work.

---

### Author Response · Authors · 2021-11-11
**Quick reply to a common question**

Dear reviewers,

Thank you for your detailed feedback from various perspectives. Reviewers Ltke and EiPP raised the same question: "Why do we focus on syn-to-real transfer?". We quickly reply to this because it is a fundamental part of our study. We will answer the rest of the questions as soon as possible.

====

An important topic of recent machine learning is studying how to build "big" models such as GPT-3 and CLIP. These models demonstrate that large pre-trained models can be extremely capable of various downstream tasks in a transfer learning setting. These big models potentially bring a critical impact to our society due to their power [1].

For vision applications, data is the main bottleneck towards building big models. The scaling law (Kaplan et al., 2020) suggests the generalization performance of a neural network is explained by three factors: data size, model size, and the amount of compute. Among these, data is the most difficult one to increase. While the last two factors "have continued to increase over the last five years, size of the largest training dataset has surprisingly remained constant" (Sun et al., 2017). This problem is decomposed into two issues.

1. The number of publicly available images is limited. Tech giants have large datasets (JFT300M at Google, Instagram dataset [2] at Facebook), but we cannot use them. Crawling WEB images is theoretically possible, but privacy and license problems exist. The realistic option is using the standard dataset such as ImageNet, but its size will not increase immediately.

2. The annotation cost is high. Annotation such as putting class labels involves humans, and it costs in terms of time and money. This is especially true for "dense" prediction tasks such as semantic segmentation and depth estimation since they require pixel-level labels and thus demand more human resources. For example, the size of Cityscape is limited to 25k. Furthermore, it is impractical for optical flow to obtain annotations, and training with synthetic images is the de facto standard [3--5].

The syn-to-real approach overcomes the two issues and thus can contribute to the foundation of big vision models. This is why we believe the syn-to-real is essential. In addition, Issues 1&2 led us to use synthetic images for pre-training (the number of real datasets is insufficient to investigate the scaling law).

====

We hope our response clarifies your concerns. If you still have questions, please reply to us. We are happy to discuss them.



- [1] Bommasani, Rishi, et al. "On the opportunities and risks of foundation models." arXiv preprint arXiv:2108.07258 (2021).
- [2] Yalniz, I. Zeki, et al. "Billion-scale semi-supervised learning for image classification." arXiv preprint arXiv:1905.00546 (2019).
- [3] Ilg, Eddy, et al. "Flownet 2.0: Evolution of optical flow estimation with deep networks." Proceedings of the IEEE conference on computer vision and pattern recognition. 2017.
- [4] Sun, D., et al. "PWC-Net: CNNs for Optical Flow Using Pyramid." Warping, and Cost Volume [J] (2017).
- [5] Jiang, Shihao, et al. "Learning to Estimate Hidden Motions with Global Motion Aggregation." arXiv preprint arXiv:2104.02409 (2021).

---

### Author Response · Authors · 2021-11-18
**Revision**

We revised the manuscript based on the reviewers' feedback. The revised parts are written in blue. The main revisions are as follows.
- Explicitly named pretraining rate \alpha and transfer gap C to avoid confusion (Section 1)
- Explained Hutter (2021) and Bahri et al. (2021) in detail (Section 2).
- Added references to transfer learning theory (Section 2)
- Corrected typo in Figure 2.
- Added explanation to parameters in Eq. (2) (Section 3.1)
- Added experiments on extrapolation (Section 4.4)

---

### Decision · Program_Chairs · 2022-01-20

**Decision:**

Reject

**Comment:**

This paper discusses an empirical scaling law in terms of samples needed for pretraining for effective downstream transfer. The reviewers liked the premise but had major concerns with the evaluation and some clarifications about empirical choices made. The paper initially received reviews tending towards rejection. The authors provided a thoughtful rebuttal that addressed some of the questions. The paper was discussed heavily and all the reviewers updated their reviews in the post-rebuttal phase. In conclusion, all reviewers still believed that their concerns regarding empirical evaluation like why evaluate only sim2real transfer, etc. still stand. AC agrees with the reviewers' consensus and encourages the authors to take the feedback into account for future submissions.